# A Little Depth Goes a Long Way: The Expressive Power of Log-Depth Transformers

## Abstract

Most analysis of transformer expressivity treats the depth (number of layers) of a model as a fixed constant, and analyzes the kinds of problems such models can solve across inputs of unbounded length. In practice, however, the context length of a trained transformer model is bounded. Thus, a more pragmatic question is: *What kinds of computation can a transformer perform on inputs of bounded length?* We formalize this by studying highly uniform transformers where the depth can grow minimally with context length. In this regime, we show that transformers with depth $O(\log C)$ can, in fact, compute solutions to two important problems for inputs bounded by some max context length $C$, namely *simulating finite automata*, which relates to the ability to track state, and *graph connectivity*, which underlies multi-step reasoning. Notably, both of these problems have previously been proven to be asymptotically beyond the reach of fixed depth transformers under standard complexity conjectures, yet empirically transformer models can successfully track state and perform multi-hop reasoning on short contexts. Our novel analysis thus explains how transformer models may rely on depth to feasibly solve problems up to bounded context that they cannot solve over long contexts. It makes actionable suggestions for practitioners as to how to minimally scale the depth of a transformer to support reasoning over long contexts, and also argues for dynamically unrolling depth as a more effective way of adding compute compared to increasing model dimension or adding a short chain of thought.

## 1 Introduction

A line of recent work has analyzed the computational power of transformers, finding that, with fixed depth, they cannot express many simple problems outside the complexity class $\mathsf{TC}^0$, including recognizing regular languages and resolving connectivity of nodes in a graph (Merrill & Sabharwal, 2023a; Chiang et al., 2023). These problems conceivably underlie many natural forms of reasoning, such as state tracking (Liu et al., 2023; Merrill et al., 2024) or resolving logical inferences across long chains (Wei et al., 2022). Thus, these results suggest inherent limitations on the types of reasoning transformer classifiers can perform. Yet, while these results establish that transformers cannot solve these problems for arbitrarily long inputs, they come with an important caveat: that transformers may still be able to solve such problems over inputs *up to some bounded length*, even if they cannot solve them exactly for inputs of arbitrary lengths. This is, in fact, aligned with a common experience that, in practice, transformer-based language models are indeed able to track state and perform multi-step reasoning successfully on small context sizes. This is analogous to how regular expressions cannot express all context-free languages, but one can write regular expressions that capture fragments of a context free language.

This perspective, coupled with the fact that *treating depth as fixed* is crucial to prior analyses placing transformers in $\mathsf{TC}^0$, motivates three related questions about depth as an important resource for a transformer, in relation to the context length over which it can express reasoning problems:

1. **Bounded Context:** Can fixed depth transformers express hard problems up to a long, but *bounded*, context length? If so, what is that bound?

2. **Dynamic Depth:** Can minimally scaling the depth of a transformer allow it to solve such problems for arbitrarily long inputs?

3. **Architecture Design:** When targeting reasoning problems like state tracking, should one add additional layers or invest test-time compute in larger model dimension, chain of thought, etc.?

We address these questions by analyzing the expressive power of "universal" transformers (also called "looped" transformers) whose depth is scaled dynamically with context length by repeating middle layers (Dehghani et al., 2019; Yang et al., 2024).[1] We capture the regime where depth grows minimally with context length by allowing the middle layers to be repeated $O(\log n)$ times. Using a universal transformer architecture allows the model to be specified using a fixed set of parameters to uniformly add layers as the context length grows. In this regime, we prove that such log-depth transformers can recognize regular languages and solve graph connectivity, two important reasoning problems shown to be beyond fixed-depth transformers in prior work (Merrill & Sabharwal, 2023a). This result has three interesting interpretations, answering the questions above:

First, this result directly shows that, by dynamically increasing their depth to $O(\log n)$, we can construct transformers that can solve regular language recognition and graph connectivity for arbitrary context length.[2]

Second, given a fixed-depth context length, a dynamic-depth transformer is a special case of a fixed-depth transformer with a uniform structure. Thus, our results show that transformers with a fixed depth $d$ can recognize regular languages and solve graph connectivity problems on inputs up to size $2^{O(d)}$, and allow us to understand how many layers are necessary. For instance, following from Theorem 1, with depth 80 (such as in LLaMA 3.1 70B), transformers can simulate finite automata on context length up to 100. Even with a depth of only 32 (such as in LLaMA 3.1 7B, OLMo 7B), they can solve graph connectivity up to a context length of 100. With depth 126 (as in LLaMA 3.1 405B), transformers can solve these problems to practically unbounded contexts. We confirm empirically that scaling depth logarithmically with a fixed context length $n$ is necessary and sufficient for learning a hard regular language recognition task.

Third, scaling depth logarithmically as a computational resource more efficiently expands the expressive power of transformers compared to scaling width (i.e., model dimension) or adding $O(\log n)$ chain-of-thought style intermediate steps (Wei et al., 2022; Nye et al., 2021). Specifically, we show that even transformers with $\text{poly}(n)$ width cannot solve the above two problems, and neither can transformers with $O(\log n)$ chain-of-thought steps.

We hope the first and third observations here will serve as actionable guidance for practitioners to choose effective model depths for reasoning over long contexts, and potentially motivate exploring the use of dynamic depth as way to efficiently introduce test-time compute for transformers.

## 2 PRELIMINARIES: UNIVERSAL TRANSFORMERS

We consider $(s, r, t)$-**universal transformers** which are defined to have $s$ fixed initial layers at the start, a sequence of $r$ layers that is repeated some number of times based on the input length, and a sequence of $t$ fixed final/terminal layers. Thus, an $(s, r, t)$-universal transformer unrolled $d(n)$ times for input length $n$ has a total of $s + rd(n) + t$ layers. A standard $d$-layer transformer is $(d, 0, 0)$-universal (equivalently, $(0, 0, d)$-universal), while a standard universal transformer (Dehghani et al., 2019; Yang et al., 2024) is $(0, 1, 0)$-universal.

**Definition 1.** A decoder-only $(s, r, t)$-universal transformer with $h$ heads, $d$ layers, model dimension $m$ (divisible by $h$), and feedforward width $w$ is specified by:

1. An embedding projection matrix $\mathbf{E}$ that maps $\mathbb{Q}^{|\Sigma|}$ to $\mathbb{Q}^m$, as well as a positional encoding function $\pi$, which we assume separates 1 from other indices (Merrill & Sabharwal, 2024);[3]
2. A list of $s$ "initial" transformer layers (defined in Section 2.1);
3. A list of $r$ "repeated" transformer layers;

---

[1]We use the term "universal" throughout because it is more standard, though "looped" is more accurate as these transformers cannot express all Turing machines with bounded precision.

[2]Following conventions in computer science, we use $\log n$ to mean $\log_2 n$.

[3]We use rationals $\mathbb{Q}$ instead of $\mathbb{R}$ so that the model has a finite description. All our simulations go through as long as at least $c \log n$ bits are used to represent rationals, similar in spirit to log-precision floats used in earlier analysis (Merrill & Sabharwal, 2023a;b).

4. A list of $t$ "final" transformer layers;

5. An unembedding projection matrix $\mathbf{U}$ that maps vectors in $\mathbb{Q}^m$ to $\mathbb{Q}^{|\Sigma|}$.

We next define how the transformer maps a sequence $w_1 \cdots w_n \in \Sigma^n$ to an output value $y \in \Sigma$; to do so, we will always specify that the transformer is **unrolled** to a specific depth function $d(n)$, which we will consider to be $d(n) = \lceil \log n \rceil$. The computation is inductively defined by the **residual stream** $\mathbf{h}_i$: a cumulative sum of all layer outputs at each token $i$. In the base case, the residual stream $\mathbf{h}_i$ is initialized to $\mathbf{h}_i^0 = \mathbf{E}(y) + \pi(i)$. We then iteratively compute $s + rd(n) + t$ more lowers, deciding which layer to use at each step as follows:

$$
L^\ell = \begin{cases}
s\text{-layer } \ell & \text{if } 1 < \ell \le s \\
r\text{-layer } (\ell - s) \bmod r & \text{if } s < \ell \le s + rd(n) \\
t\text{-layer } \ell - s - rd(n) & \text{otherwise.}
\end{cases}
$$

We then compute $\mathbf{h}_1^\ell, \ldots, \mathbf{h}_n^\ell = L^\ell(\mathbf{h}_1^{\ell-1}, \ldots, \mathbf{h}_n^{\ell-1})$. Finally, the output of the transformer is a token determined by first computing the logits $\mathbf{h}_n^{\ell-1}\mathbf{U}$ and selecting token with maximum score. We can identify special tokens in $\Sigma$ with "accept" and "reject" and define a transformer to **recognize** a language $L$ if, for every $w \in \Sigma^*$, it outputs "accept" if $w \in L$ and "reject" otherwise.

An $(s, r, t)$-transformer unrolled to some fixed depth can be viewed as a "uniform" special case of a fixed-depth transformer. Thus, constructions of dynamic-depth transformers (depth $d(n)$ for inputs of length $n$) imply that, given any bounded context length $N$, there also exists a fixed-depth transformer with depth $d(N)$ for the task at hand. The fact that this can be done with a looped transformer with dynamic depth is, in fact, a stronger condition that shows the construction is uniform, which is formally important as non-uniform models of computation can have very strong and unrealistic power (cf. Merrill et al., 2022a). In this way, our results about looped transformers will provide insights about standard, non-looped transformers with bounded context lengths.

## 2.1 TRANSFORMER SUBLAYERS

To make Definition 1 well-defined, we will next describe the structure of the self-attention and feedforward sublayers that make up the structure of each transformer layer. Our definition of the transformer will have two minor differences from practice:

1. **Averaging-hard attention** (a.k.a., saturated attention): attention weight is split uniformly across the tokens with maximum attention scores.

2. **Masked pre-norm**: We assume standard pre-norm (Xiong et al., 2020) but add a learned mask vector that can select specific dimensions of the residual stream for each layer's input.

Each sublayer will take as input a sequence of normalized residual stream values:

$$
\mathbf{z}_i = \mathsf{layer\_norm}(\mathbf{m}\mathbf{h}_i),
$$

where layer-norm can be standard layer-norm (Ba et al., 2016) or RMS norm (Zhang & Sennrich, 2019). The sublayer then maps $\mathbf{z}_1, \ldots, \mathbf{z}_n$ to a sequence of updates to the residual stream $\delta_1, \ldots, \delta_n$, and the residual stream is updated as $\mathbf{h}_i' = \mathbf{h}_i + \delta_i$.

**Definition 2** (Self-attention sublayer). The self-attention sublayer is parameterized by a mask $\mathbf{m} \in \mathbb{Q}^m$, output projection matrix $\mathbf{W} \in \mathbb{Q}^{m \times m}$, and, for $1 \le k \le h$, query, key, and value matrices $\mathbf{Q}^k \in \mathbb{Q}^{m \times (m/h)}, \mathbf{K}^k \in \mathbb{Q}^{m \times (m/h)}, \mathbf{V}^k \in \mathbb{Q}^{m \times (m/h)}$.

Given its input $\mathbf{z}_i$, the self-attention sublayer computes queries $\mathbf{q}_i = \mathbf{z}_i\mathbf{Q}^k$, keys $\mathbf{k}_i = \mathbf{z}_i\mathbf{K}^k$, and values $\mathbf{v}_i = \mathbf{z}_i\mathbf{V}^k$. Next, these values are used to compute the attention head outputs:

$$
\mathbf{a}_{i,k} = \lim_{\alpha \to \infty} \sum_{j=1}^c \frac{\exp(\alpha\mathbf{q}_{i,k}\mathbf{k}_{j,k})}{Z_{i,k}} \cdot \mathbf{v}_{j,k}, \text{ where } Z_{i,k} = \sum_{j=1}^c \exp\left(\alpha\mathbf{q}_{i,k}\mathbf{k}_{j,k}\right)
$$

where $c = i$ for causal attention and $c = n$ for unmasked attention. Attention is made saturated to focus on the argmax positions (through the $\alpha$ limit). Finally, the attention heads are aggregated to create an output to the residual stream:

$$
\delta_i = \mathsf{concat}(\mathbf{a}_{i,1}, \ldots, \mathbf{a}_{i,h}) \cdot \mathbf{W}.
$$

**Definition 3** (Feedforward sublayer). The feedforward sublayer at layer $\ell$ is parameterized by a mask $\mathbf{m} \in \mathbb{Q}^m$ and projections $\mathbf{W} \in \mathbb{Q}^{m \times w}$ and $\mathbf{U} \in \mathbb{Q}^{w \times m}$.

A feedforward layer computes a local update to the residual stream according to

$$\delta_i = \mathsf{ReLU}(\mathbf{z}_i \mathbf{W}) \mathbf{U}.$$

## 2.2 Memory Management in Universal Transformers

A technical challenge when working with universal transformers that add values to the residual stream is that if one is not careful, outputs from the previous iteration of a layer may interfere with its computation at a later iteration. This necessitates "memory management" of individual cells in which the transformer stores values. In particular, any intermediate values stored by a layer must be "reset" to 0 and any desired output values must be correctly updated after use in subsequent layers.

Appendix A discusses in detail how values in $\{-1, 0, 1\}$ can be stored directly in the residual stream, while a general scalar $z$ can be stored either as $\psi(z) = \langle z, 1, -z, -1 \rangle$ in its *unnormalized form* or as the unit vector $\phi(z) = \psi(z)/\sqrt{z^2 + 1}$ in its *normalized form*. Importantly, whichever way a general $z$ is stored, when it is read using masked pre-norm, we obtain $\phi(z)$. Thus, if $\psi(z)$ is stored as an intermediate output, resetting the corresponding residual stream cells in the next layer will often require recomputing $\psi(z)$ again in the next layer and adding $-\psi(z)$ to those cells to reset their value to 0. We will use a similar mechanism to reset or update a scalar added to a single cell of the residual stream, such as in the proof of Lemma 5. Further details are deferred to Appendix A.

## 3 Fixed Depth Transformers Can Divide Small Integers

A useful primitive for coordinating information routing in a log-depth transformer will be dividing integers and computing remainders. We therefore start by proving that transformers can perform integer division for small numbers, which will be a useful tool for our main results. Specifically, we show that given a non-negative integer $a_i$ no larger than the current position $i$, one can compute and store the (normalized) quotient and remainder when $a_i$ is divided by an integer $m$. This effectively means transformers can perform arithmetic modulo $m$ for small integers.

We note that there are some high-level similarities between our division construction and a modular counting construction from Strobl et al. (2024), though the tools (and simplifying assumptions) used by each are different. Specifically, their approach relies on nonstandard position embeddings whereas ours makes heavy use of masked pre-norm.

**Lemma 1.** *Let $a_i, b_i, c_i, m \in \mathbb{Z}^{\geq 0}$ be such that $a_i = b_i m + c_i$ where $a_i \leq i$ and $c_i < m$. Suppose $\psi(i)$, $\psi(m)$, and $\phi(a_i)$ (or $\psi(a_i)$) are present in the residual stream of a transformer at each token $i$. Then, there exists a 7-layer transformer with causally masked attention and masked pre-norm that, on any input sequence, adds $\phi(b_i)$ and $\phi(c_i)$ to the residual stream at each token $i$.*

*Proof.* The overall idea is as follows. In the first layer, each position $i$ outputs an indicator of whether it's a multiple of $m$. It also adds $\phi(j)$ to the residual stream such that $j$ is the quotient $i/m$ if $i$ is a multiple of $m$. In the second layer, each position $i$ attends to the nearest position $j \leq i$ that is a multiple of $m$ and retrieves the (normalized) quotient stored there, which is $j/m = \lfloor i/m \rfloor$. It adds this (normalized) quotient in its own residual stream. We then use Lemma 4 to construct a third layer that adds $\phi(i-1)$ and $\phi(i-2)$ to the residual stream. A fourth layer checks in parallel whether the quotient stored at $i$ matches the quotients stored at $i-1$ and $i-2$, respectively. In the fifth layer, position $i$ counts the number of positions storing the same quotient as $i$, excluding the first such position. Finally, in the sixth layer, position $i$ attends to position $a_i$ to compute and add to the residual stream $\phi(\lfloor a_i/m \rfloor)$ (which is $\phi(b_i)$) and $\phi(a_i - m\lfloor a_i/m \rfloor)$ (which is $\phi(c_i)$). We next describe a detailed implementation of the construction, followed by an argument of its correctness.

Construction. The first layer uses an attention head with queries, keys, and values computed as follows. The query at position $i$ is $q_i = \phi(i, m) = \phi(i/m)$ computed via Lemma 2 leveraging the assumption that $\psi(i)$ and $\psi(m)$ are present in the residual stream. The key and value at position $j$ are $k_j = v_j = \phi(j)$. Let $h_i^1 = \phi(j)$ denote the head's output. The layer adds $h_i^1$ to the residual stream and also adds $e_i = \mathbb{I}(h_i^1 = \phi(i/m))$ using Lemma 5 (scalar equality check) on the first

coordinate of $h_i^1$ and $\phi(i/m)$. As we will argue below, this layer has the intended behavior: $e_i = 1$ if and only if $i$ is a multiple of $m$ and, if $e_i = 1$, then the value it stores in the residual stream via $h_i^1$ is precisely the (normalized) quotient $i/m$.[4]

The second layer uses a head that attends with query $q_i = \langle 1, 1 \rangle$, key $k_j = \langle e_j, [\phi(j)]_0 \rangle$, and value $v_j = h_j^1$; note that both $e_j$ and $h_j^1$ can be read from the residual stream using masked pre-norm. This head attends to all positions $j \leq i$ that are multiples of $m$ (where $e_j = 1$), with $[\phi(j)]_0$, the first component of $\phi(j)$, serving as a tie-breaking term for breaking ties in favor of the *nearest* multiple of $m$. Let $h_i^2 = h_j^1$ denote the head's output. The layer adds $h_i^2$ to the residual stream at position $i$. As we will argue below, $h_i^2 = \phi(j/m)$ where $j/m$ is precisely the quotient stored in the residual stream at the multiple $j$ of $m$ that is closest to (and no larger than) $i$, which by definition is $\lfloor i/m \rfloor$. The layer thus adds $\phi(\lfloor i/m \rfloor)$ to the residual stream at position $i$.

The third layer uses Lemma 4 to add $\phi(i-1)$ and $\phi(i-2)$ to the residual stream at $i$.

In parallel for $k \in \{1, 2\}$, the fourth layer attends with query $q_i = \phi(i - k)$, key $k_j = \phi(j)$, and value $v_j = \phi(\lfloor j/m \rfloor)$ to retrieve the quotient stored at position $i - k$. It uses Lemma 5 (on the first coordinate) to store in the residual stream a boolean $b_i^k = \mathbb{I}(\phi(\lfloor i/m \rfloor) = \phi(\lfloor (i-k)/m \rfloor))$, indicating whether the quotient stored at $i$ matches the quotient stored at $i - k$.

In the fifth layer, position $i$ attends with query $q_i = \langle \phi(\lfloor i/m \rfloor), 1 \rangle$, key $k_j = \langle \phi(\lfloor j/m \rfloor), b_j^1 \rangle$, and value $v_j = 1 - b_j^2$; note that $b_i^k$ can be retrieved from the residual stream. This head thus attends to every position with the same quotient as the current token besides the initial such position, with value 1 at the second such token and 0 elsewhere. Assuming $m$ does not divide $i$, this head will attend to precisely $i - m\lfloor i/m \rfloor$ positions and return $f_i = 1/(i - m\lfloor i/m \rfloor)$ as the head output. The layer adds the vector $\psi(1, f_i)$ defined as $\langle 1, f_i, -1, -f_i \rangle$ to the residual stream at position $i$. This, when read in the next layer using masked pre-norm, will yield $\phi(1, f_i) = \phi(1/f_i)$. On the other hand, if $m$ does divide $i$ (which can be checked with a separate, parallel head), we write $\psi(0)$ to the residual stream, which, when read by the next layer, will yield $\phi(0)$.

The sixth layer attends with query $q_i = \phi(a_i)$, key $k_j = \phi(j)$, and value $v_j = \langle h_j^2, \phi(1/f_j) \rangle$. Recall that $\phi(1/f_j)$ can be read from the residual stream as discussed above. Further, the layer can recompute $f_j$ (or 0 in case $m$ divides $i$) and write $-\psi(1, f_j)$ (or $-\psi(0)$, respectively) to the same coordinates, thereby resetting those cells to 0. Since $a_i \leq i$, the query matches exactly one position $j = a_i$, and the head retrieves $\langle h_{a_i}^2, 1/\phi(1/f_{a_i}) \rangle$. This, by construction, is $\langle \phi(\lfloor a_i/m \rfloor), \phi(i - m\lfloor a_i/m \rfloor) \rangle$, which equals $\langle \phi(b_i), \phi(c_i) \rangle$. The layer can thus store $\phi(b_i)$ and $\phi(c_i)$ to the residual stream at position $i$, as desired.

The seventh and final layer cleans up any remaining intermediate values stored in the residual stream, setting them back to 0 as per Lemma 5. This is possible because all values $v$ are of the form $\phi(x)$ or boolean, so adding $-\phi(v)$ will reset the cell to 0.

Correctness. We now argue that each layer, as constructed above, conforms to its intended behavior.

In the first layer, suppose first that $i$ is a multiple of $m$. In this case, there exists a position $j^* \leq i$ such that $i = mj^*$, which means the query $q_i = \phi(i/m) = \phi(j^*)$ exactly matches the key $k_{j^*}$. The head will thus return $v_{j^*} = \phi(j^*) = \phi(i/m)$, representing precisely the quotient $i/m$. Further, the equality check will pass, making $e_i = 1$. The layer thus behaves as intended when $i$ is a multiple of $m$. On the other hand, when $i$ is *not* a multiple of $m$, no such $j^*$ exists. The head will instead attend to some $j$ for which $i \neq mj$ and therefore $\phi(i/m) \neq \phi(j)$, making the subsequent equality check fail and setting $e_i = 0$, as intended.

In the second layer, $q_i \cdot k_j = e_j - [\phi(j)]_0$ where $[\phi(j)]_0 = j/\sqrt{2j^2 + 2}$ is the first coordinate of $\phi(j)$. Note that $[\phi(j)]_0 \in [0, 1)$ for positions $j \leq i$ and that it is monotonically increasing in $j$. It follows that the dot product is maximized at the largest $j \leq i$ such that $e_j = 1$, i.e., at the largest $j \leq i$ that is a multiple of $m$. This $j$ has the property that $\lfloor i/m \rfloor = j/m$. Thus, the head at this layer attends solely to this $j$ and retrieves the value $\phi(j/m) = \phi(\lfloor i/m \rfloor)$ as intended.

The correctness of the third and fourth layer is easy to verify.

---

[4]As described in Lemma 5, a component will be added to the second layer to reset intermediate memory cells used in the first layer to 0 (this will happen analogously in later layers, but we will omit mentioning it).

In the fifth layer, $q_i \cdot k_j \leq 2$ and the dot product achieves this upper limit exactly when two conditions hold: $b_j^1 = 1$ and $\lfloor i/m \rfloor = \lfloor j/m \rfloor$. Thus, as desired, the head at $i$ attends to all positions $j \leq i$ that have the same quotient as $i$ and also have $b_j^1 = 1$. Write $i$ as $i = b'm + c'$ for some $c' < m$. It follows that the query-key dot product is maximized precisely at the $c'$ positions $b'm + 1, b'm + 2, \ldots, b'm + c'$. Of these positions, only $b'm + 1$ has the property that the quotient there is *not* the same as the quotient two position earlier, as captured by the value $v_j = 1 - b_j^2$. Thus, the value $v_j$ is 1 among these positions only at $j = b'm + 1$, and 0 elsewhere. The head thus attends uniformly at $c'$ positions and retrieves $1/c'$. By construction, $c' = i - b'm = i - \lfloor i/m \rfloor m$, showing that this layer also behaves as intended.

Finally, that the sixth and seventh layers operate as desired is easy to see from the construction. $\square$

## 4   Log Depth Enables Recognizing Regular Languages

One natural problem that constant-depth transformers cannot express is recognizing regular languages, which is closely related to state tracking (Liu et al., 2023; Merrill et al., 2024). Liu et al. (2023, Theorem 1) show how a log-depth transformer can recognize regular languages using a binary tree construction similar to associative scan (Hillis & Steele Jr, 1986). However, their result requires simplifying assumptions, removing residual connections from the transformer and assuming specific positional encodings. As discussed in Section 2.2, dealing with residual connections is particularly tricky in universal transformers, requiring proper memory management of cells in the residual stream so that outputs from the previous iteration of a layer interfere with a later iteration. Our result therefore refines that of Liu et al. (2023) to hold with a more general universal transformer model that uses residual connections and does not rely on specific positional encodings:

**Theorem 1.** *Let $L$ be a regular language over $\Sigma$ and $\$ \notin \Sigma$. Then there exists a $(0, 7, 9)$-universal transformer with causal masking that, on any string $w\$$, recognizes whether $w \in L$ when unrolled to $\lceil \log_2 |w| \rceil$ depth.*

*Proof.* Regular language recognition can be framed as multiplying a sequence of elements in the automaton's transition monoid (Myhill, 1957; Thérien, 1981). It thus suffices to show how elements in a finite monoid can be multiplied with log depth. We show how a log-depth universal transformer can implement the standard binary tree construction (Barrington & Thérien, 1988; Liu et al., 2023; Merrill et al., 2024) where each level multiplies two items, meaning the overall depth is $O(\log |w|)$. We will represent a tree over the input tokens within the transformer. Each level of the tree will take 5 transformer layers. We define a notion of active tokens: at level 0, all tokens are active, and, at level $\ell$, tokens at $t \cdot 2^\ell - 1$ for any $t$ will remain active, and all other tokens will be marked as inactive. As an invariant, active token $i = t \cdot 2^\ell - 1$ in level $\ell$ will store a unit-norm vector $\delta_i^\ell$ that represents the cumulative product of tokens from $i - 2^\ell + 1$ to $i$.

We now proceed by induction over $\ell$, defining the behavior of non-$\$$ tokens at layers that make up level $\ell$. The current group element $\delta_i^\ell$ stored at active token $i$ is, by inductive assumption, the cumulative product from $i - 2^\ell + 1$ to $i$. Let $\alpha_i^\ell$ denote that token $i$ is active. By Lemma 4 we use a layer to store $i - 1$ at token $i$. The next layer attends with query $\phi(i - 1)$, key $\phi(j)$, and value $\delta_j^\ell$ to retrieve $\delta_{i-1}^\ell$, the group element stored at the previous token. Finally, another layer attends with query $\vec{1}$, key $\langle \phi(j)_1, \alpha_i^\ell \rangle$, and value $\delta_{j-1}^\ell$ to retrieve the group element $\delta_{j*}^\ell$ stored at the previous active token, which represents the cumulative product from $i - 2 \cdot 2^\ell + 1$ to $i - 2^\ell$. Next, we will use two layers to update $\delta_i^\ell \leftarrow \delta_i^{\ell+1}$ and $\delta_j^\ell \leftarrow \vec{0}$, which is achieved as follows. First, we assert there exists a single feedforward layer that uses a table lookup to compute $\delta_{j*}^\ell, \delta_i^\ell \mapsto d$ such that

$$\frac{d}{\|d\|} = \delta_{j*}^\ell \cdot \delta_i^\ell = \delta_i^{\ell+1}.$$

Next, we invoke Lemma 3 to construct a layer that adds $d$ to an empty cell of the residual stream and then another layer that deletes it. This second layer can now read both $\delta_i^\ell, \delta_{j*}^\ell$ and $\delta_i^{\ell+1}$ (from $d$) as input, and we modify it to add $\delta_i^{\ell+1} - \delta_i^\ell$ to $\delta_i^\ell$, changing its value to $\delta_i^{\ell+1}$. Similarly, we modify it to add $-\delta_{j*}^\ell$ to $\delta_{j*}^\ell$ to set it to 0. A feedforward network then subtracts $\delta_i^\ell$ from the residual stream and adds $\delta_i^\ell \cdot \delta_j^\ell$. This requires at most 4 layers.

To determine activeness in layer $\ell + 1$, each token $i$ attends to its left to compute $c_i/i$, where $c_i$ is the prefix count of active tokens, inclusive of the current token. We then compute $\phi(c_i/i, 1/i) = \phi(c_i)$ and store $c_i$ it temporarily in the residual stream. At this point, we use Lemma 1 to construct 7 layers that compute $c_i \bmod 2$ with no storage overhead. The current token is marked as active in layer $\ell + 1$ iff $c_i = 0 \mod 2$, which is equivalent to checking whether $i = t \cdot 2^\ell - 1$ for some $t$. In addition to updating the new activeness $\alpha_i^{\ell+1}$, we also persist store the previous activeness $\alpha_i^\ell$ in a separate cell of the residual stream and clear $c_i$. This requires at most 8 layers.

Finally, we describe how to aggregate the cumulative product at the $\$$ token, which happens in parallel to the behavior at other tokens. Let $\delta_\$^\ell$ be a monoid element stored at $\$$ that is initialized to the identity and will be updated at each layer. Using the previously stored value $i - 1$, we can use a layer to compute and store $\alpha_{i-1}^\ell$ and $\alpha_{i-1}^{\ell+1}$ at each $i$. A head then attends with query $\vec{1}$, key $\langle \phi(j)_1, 10 \cdot \alpha_{i-1}^\ell \rangle$, and value $\langle (1 - \alpha_{j-1}^{\ell+1}) \cdot \delta_{j-1}^{\ell+1} \rangle$. This retrieves a value from the previous active token $j$ at level $\ell$ that is $\delta_j^\ell$ if $j$ will become inactive at $\ell + 1$ and $\vec{0}$ otherwise. Iff $\delta_j^\ell$ is retrieved, a feedforward network subtracts $\delta_\$^\ell$ from the residual stream and adds $\delta_j^\ell \cdot \delta_\$^\ell$. This guarantees that whenever a tree is deactivated, its cumulative product is incorporated into $\delta_\$^\ell$. Thus, after $\ell = \lceil \log_2 |w| \rceil + 1$ levels, $\delta_\$^\ell$ will be the transition monoid element for $w$. We can use one additional layer to check whether this monoid element maps the initial state to an accepting state using a finite lookup table. Overall, this can be expressed with 8 layers repeated $\lceil \log_2 |w| \rceil$ times and 9 final layers (to implement the additional step beyond $\lceil \log n \rceil$). $\qquad \square$

Theorem 1 thus reveals that running a transformer to $\log n$ depth on inputs of length $n$ unlocks new power compared to a fixed-depth transformer.

**Remark.** The idea of this theorem can be generalized beyond regular languages: if a $c$ layer transformer can perform some binary associative operation $\oplus$, then one can construct an $\mathrm{O}(c \log n)$ layer transformer that computes the iterated version of the operator on $n$ values, $x_1 \oplus x_2 \oplus \ldots \oplus x_n$. One natural iterated problem is **iterated matrix multiplication**. If the matrices come from a fixed set (e.g., they are fixed size $k \times k$ matrices over booleans), then our result for regular languages shows that this task can be performed. However, if the matrices are not from a fixed set (e.g., they contain general integer or rational values, or the matrix itself is of size $n \times n$), then it is unclear whether log-depth transformers can solve the iterated multiplication problem; in fact, for $n \times n$ integer matrices, it is unknown whether they can even compute binary multiplication.

## 5 LOG DEPTH ENABLES GRAPH CONNECTIVITY

In the **graph connectivity problem**, the input is a graph $G$, along with a source vertex $s$ and a target vertex $t$. The task is to determine whether $G$ has a path from $s$ to $t$. This is a core problem at the heart of many computational questions in areas as diverse as network security, routing and navigation, chip design, and—perhaps most commonly for language models—multi-step reasoning. This problem is known to be complete for the class of logspace Turing machines (Reingold, 2008; Immerman, 1998), which means that, under common complexity theory beliefs, it cannot be solved accurately by fixed-depth transformer encoders, which can only solve problems in the smaller class $\mathsf{TC}^0$. In fact, it is believed to not be solvable even with log-depth AND/OR circuits ($\mathsf{NC}^1$). However, logspace Turing machines can be simulated by log-depth *threshold* circuits ($\mathsf{TC}^1$) (Barrington & Maciel, 2000), which opens up a natural question: *Can log-depth transformers, which are in $\mathsf{TC}^1$, solve graph connectivity?* We show in this section that the answer is yes.

**Theorem 2.** *There exists an $(17, 2, 1)$-universal transformer $T$ with both causal and unmasked heads that, when unrolled $\lceil \log_2 n \rceil$ times, solves the connectivity problem on (directed or undirected) graphs over $n$ vertices: given as input the $n \times n$ adjacency matrix of a graph $G$, $n^3$ padding tokens, and $s, t \in \{1, \ldots n\}$ in unary notation, $T$ determines whether $G$ has a path from vertex $s$ to vertex $t$.*

*Proof.* In Appendix B. $\qquad \square$

# 6 THE RELATIVE EFFICIENCY OF GROWING DEPTH, GROWING WIDTH, AND CHAIN OF THOUGHT

We now consider how increasing the depth compares to other methods of extending the computational resources that a transformer can perform. One natural question is how increasing depth compares to increasing width: it turns out that, whereas slightly increasing depth expands expressive power beyond $\mathsf{TC}^0$, doing the same by increasing width would require width to grow *superpolynomially* with sequence length, which is infeasible. Another natural comparison is between increasing depth and adding chain-of-thought (CoT) steps, as both are ways to expand the test-time compute avaiable to a pretrained model. We now draw on related results in the literature to compare the efficiency of growing depth to growing width or adding chain of thought.

## 6.1 WIDE TRANSFORMERS WITH FIXED DEPTH REMAIN IN $\mathsf{TC}^0$

We have shown that growing the transformer's depth minimally allows it to express key problems that are likely outside $\mathsf{TC}^0$. Does growing the width of the model have the same effect? How does increasing the width of a model change its expressive power compared to increasing the depth? We draw on related results in the literature to compare the efficiency of growing depth to growing width or adding chain of thought. While we have shown that growing depth logarithmically enables solving some problems outside $\mathsf{TC}^0$, width must be scaled *exponentially* with $n$ to make problems outside $\mathsf{TC}^0$ expressible over sequences up to length $n$.

**Theorem 3.** *Consider a transformer with fixed depth whose width (model dimension) grows as a polynomial of $n$ and whose weights on input length $n$ (to accomodate growing width) are computable in* $\mathsf{L}$. *Then this transformer can be simulated in* $\mathsf{L}$-*uniform* $\mathsf{TC}^0$.

*Proof.* The proof is a straightforward extension of Theorem 1 in Merrill & Sabharwal (2023a). For completeness, we give a proof sketch in Appendix C. □

This result shows that, to solve reasoning problems outside $\mathsf{TC}^0$ over a context length $n$, growing depth is much more efficient than growing width. Of course, there may be other types of problems (e.g., those that are knowledge intensive are very parallelizable) where growing width might be more important than growing depth. See Petty et al. (2024) for an empirical investigation of the efficiency of scaling depth vs. width on language modeling, semantic parsing, and other tasks.

## 6.2 TRANSFORMERS WITH LOG CHAIN-OF-THOUGHT STEPS REMAIN IN $\mathsf{TC}^0$

Merrill & Sabharwal (2024) analyze the power of transformers with $O(\log n)$ chain-of-thought steps, showing it is at most $\mathsf{L}$. However, we have shown that transformers with $O(\log n)$ depth can solve directed graph connectivity, which is NL-complete: this suggests growing depth has some power beyond growing chain of thought unless $\mathsf{L} = \mathsf{NL}$. In fact, this can be extended (Li et al., 2024) to show transformers with $O(\log n)$ chain of thought cannot solve *any* problem outside $\mathsf{TC}^0$.

**Theorem 4.** *Any language recognized by a transformer with $O(\log n)$ steps of chain of thought (cf. Merrill & Sabharwal, 2024) is in* $\mathsf{TC}^0$.

*Proof.* This follows from the hierarchy in Figure 10 of Li et al. (2024). For completeness, we give a proof sketch in Appendix C. □

Thus, while giving a model $O(\log n)$ steps of chain of thought does not increase its expressive power beyond $\mathsf{TC}^0$, our Theorems 1 and 2 allow $\Theta(\log n)$ to solve key problems that are (likely) outside $\mathsf{TC}^0$. This demonstrates an advantage of dynamic depth over chain of thought as a form of test-time compute in a specific case. In the future, it would be interesting to explore this comparison for generally across other types of problems.

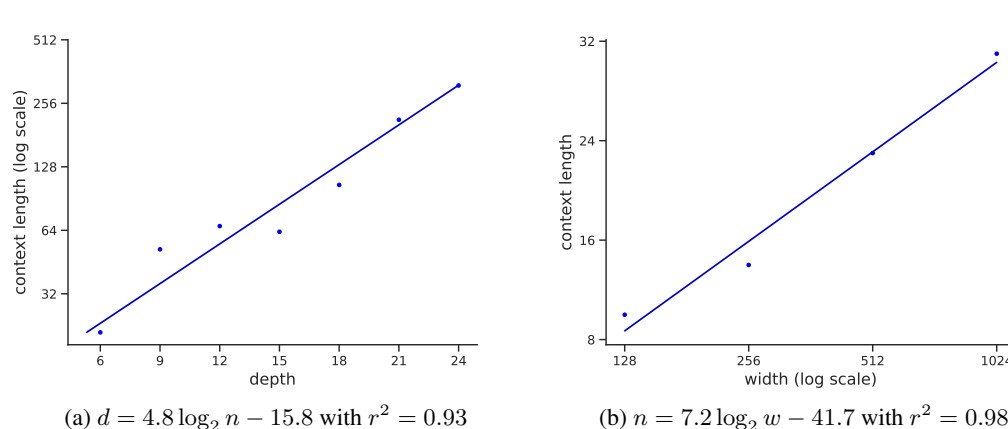

(a) $d = 4.8 \log_2 n - 15.8$ with $r^2 = 0.93$      (b) $n = 7.2 \log_2 w - 41.7$ with $r^2 = 0.98$

Figure 1: Strong linear fits imply theory/experiment match for modeling the impact of depth and width on effective context length for the $A_5$ state tracking task, a canonical hard regular language recognition problem. As predicted by Theorem 1 and Theorem 3, to recognize strings of length $n$, depth only needs to increase minimally $\propto \log n$ while width must increase drastically $\propto \exp(\Theta(n))$.

# 7 EMPIRICAL VALIDATION OF PREDICTED DEPTH AND WIDTH SCALING

Theorems 1 and 3 Our theory makes empirically testable predictions about the relationship between a model's depth (and width) and the effective context length for key reasoning problems outside $\mathsf{TC}^0$. Specifically, as predicted by Theorem 1, is it the case that recognizing regular languages over strings of length $n$ requires depth $\propto \log n$? On the other hand, as predicted by Theorem 3, must the width scale $\propto \exp(\Theta(n))$ in order to recognize strings of length $n$? Finally, if these relationships hold, can we empirically quantify the constant factors?

We ran an extensive set of experiments to address these questions, training models of different depths and widths on the $A_5$ state tracking task (Merrill et al., 2024), which is a canonical testbed for hard regular language recognition (Theorem 1). The input to the task is a sequence of elements in $A_5$ (the group of even permutations over 5 elements), and the label at each token is the cumulative product of previous permutations up to and including that token (which is itself an element of $A_5$).

We train several (non-universal) transformers with the same architecture used by Merrill et al. (2024) on 100 million sequences of the $A_5$ task of varying lengths up to 1024 (this took 1000 GPU hours). In order to understand the impact of depth and width in a controlled way, we train two series of transformers: the first with width fixed to 512 and depth varying in $\{6, 9.12, 15, 18, 21, 24\}$, and the second with depth fixed to 6 and width varying in $\{128, 256, 512, 1024\}$. After each model is trained, we measure accuracy at each token index from 1 to 1024 and define $n^*$ as the maximum token index at which the model achieved at least 95% validation accuracy. As we trained several seeds with the same depth and width, we aggregate these results across all models with the same depth and width by taking the best-performing (max $n^*$) model. We can then plot $n^*$, which represents the effective context length up to which a model can solve the $A_5$ problem, as a function of either depth or width, holding the other variable fixed. We then evaluate if the predicted theoretical relationships between depth, width, and context length hold via an $r^2$ statistic.

The results are shown in Figure 1. When varying depth (Figure 1a), there is a very strong positive correlation ($r^2 = 0.93$) between effective context length depth (x-axis) and $\log n^*$ (y-axis, log scale). When varying width (Figure 1b), there is an even stronger positive correlation ($r^2 = 0.98$) between log width (x-axis, log scale) and $n^*$ (y-axis). These results provide strong empirical support for our theoretical predictions that, to recognize regular languages over strings of length $n$, increasing depth logarithmically in $n$ will suffice (Theorem 1), but depth must increase exponentially in $n$ (Theorem 3). Figure 1 also give us a strongly predictive functional form to quantify the impact of scaling depth or width on the effective context length for regular language recognition. The empirical slope for the depth relationship is is 4.8 layers per log tokens, which is more compact then the slope of 7 in Theorem 1. As our construction was not fully tight, future work could refine it towards the

slope found in practice. Overall, these empirical results show that, in practice, the impact of depth and width on effective context length for regular language recognition is as predicted by our theory.

## 8 LIMITATIONS OF LOG DEPTH

We have shown that increasing transformer depth logarithmically with the input sequence length allows transformers to solve some problems they cannot solve with constant depth, under standard conjectures. Is logarithmic depth sufficient for transformers to solve any inherently sequential problem, or are there some problems that cannot be made solvable in this way?

It turns out there are many problems that likely are not made expressible by log depth. We know that log-depth transformers can be simulated in $\mathsf{TC}^1$. Thus, unless $\mathsf{NC} = \mathsf{P}$, log-depth (or even *poly-log* depth, i.e., $\log^k n$) transformers cannot express P-complete problems including solving linear equalities, in-context context-free language recognition (given both a grammar $G$ and string $x$ as input, does $G$ generate $x$?), circuit evaluation, and determining the satisfiability of Horn clauses. In future work, it would be interesting to empirically test whether solving these problems over contexts of length $n$ requires $\Theta(n)$ or $\mathrm{poly}(n)$ depth in practice.

Beyond P-complete problems, it is conceivable that other natural reasoning problems could be inexpressible by log-depth transformers. Interesting candidates include context-free recognition (generalizing regular languages; Theorem 1), which is in $\mathsf{NC}^2$ (Ruzzo, 1981). An even simpler problem where we do not have a log-depth transformer construction (but which is in $\mathsf{NC}^1$) is boolean formula evaluation. In future work, it would be interesting to further study the depth required for these problems and identify separations between transformers with $\Theta(\log n)$ and $\Theta(\log^2 n)$ depth, which we believe may correspond roughly to a boundary for what is efficient to train in practice. Further theoretical analysis and depth scaling experiments on tasks like context-free recognition could improve our understanding of where the exact upper frontier for log-depth transformers lies.

## 9 CONCLUSION

We have shown that recognizing regular languages and graph connectivity, two key problems inexpressible by fixed-depth transformers, become expressible if the depth of the transformer can grow *very slightly* (logarithmically) with the context length. This implies that transformers with fixed depth $d$ can solve these problems up to context length at least $2^{O(d)}$. Thus, while these problems are not solvable in general by fixed-depth transformers, our results reveal that one only has to minimally scale depth to make them expressible up to some bounded context length. Further, we showed that scaling depth to solve these problems is more efficient than scaling width (which requires superpolynomial increase) or scaling chain-of-thought steps (which requires more than logarithmic increase). In future work, it would thus be interesting to explore whether universal transformers can realize this theoretical efficiency in practice to provide more efficient long-context reasoning than chain of thought prompting.

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

## A   BUILDING BLOCKS

### A.1   RESIDUAL STREAM STORAGE INTERFACE

Our masked pre-norm transformer architecture always normalizes values when reading them from the residual stream. This means that it's not always the case that what's added to the residual stream by one layer is accessible as-is in future layers, which can be problematic if there is a need to "erase" that value. We discuss how values are stored and, if needed, erased from the stream.

For any general scalar $z$, storing $z$ in the residual stream results in $\text{sgn}(z)$ being retrieved when masked pre-norm is applied to this cell. This will be useful when we want to collapse multiple values or perform equality or threshold checks. As a special case, when $z \in \{-1, 0, 1\}$, the retrieved value after masked pre-norm is precisely $z$. Thus scalars in $\{-1, 0, 1\}$ can be stored and retrieved without any information loss.

When a general scalar $z$ needs to be preserved, we store it as a 4-dimensional vector. Let $\psi(z) = \langle z, 1, -z, -1 \rangle$ be its *unnormalized representation* and the corresponding $0$-centered unit vector $\phi(z) = \psi(z)/\sqrt{z^2 + 1}$ be its *normalized representation*. We say that a scalar $z$ is **stored in the residual stream** if some set of four indices contain either $\psi(z)$ or $\phi(z)$. Note that a masked pre-norm applied to the positions containing $\psi(z)$ or $\phi(z)$ yields $\phi(z)$. Thus, once a scalar $z$ is stored in the residual stream in either form, it remains available in subsequent layers as $\phi(z)$. We will write "a transformer layer **stores** $z$" to mean it adds either $\psi(z)$ or $\phi(z)$ to the residual stream, depending on which one it has immediate access to.

Individual scalars stored in the residual stream can be trivially retrieved by masked pre-norm. In addition, the hashes of pairs of stored scalars can be easily retrieved as well:

**Lemma 2.** *Let* $\langle x_1, y_1 \rangle, \dots, \langle x_k, y_k \rangle$ *be pairs of integers stored in the residual stream. There exists a masked pre-norm that computes* $\langle \phi(x_1, y_1), \dots, \phi(x_k, y_k) \rangle$ *or, equivalently,* $\langle \phi(x_1/y_1), \dots, \phi(x_k/y_k) \rangle$.

*Proof.* We apply a masked pre-norm to the positions where $x_1, \dots, x_k$ and $y_1, \dots, y_k$ are stored:

$$\frac{1}{\sqrt{2k}} \langle \phi(x_1, y_1), \dots, \phi(x_k, y_k) \rangle.$$

We can hardcode the scalar multiplier of the layer-norm output to remove the scalar factor (or equivalently, bake it into the next linear transformation). $\square$

In the repeated layers of a universal transformer, we will want need overwrite the value stored in a particular register of the residual stream with a new value. That is, given $x_\ell$ is stored at layer $\ell$, we will want to store some new value $x_{\ell+1}$ instead. In most cases, this will involve computing some intermediate values and then removing them from the residual stream. The following lemma turns out to be useful for constructions of this form:

**Lemma 3.** *Assume there exists a single transformer layer that writes an update $\delta_i$ to the residual stream $\mathbf{h}_i$ using indices at which $\delta_i$ is 0. Then there is a block of two transformer layers that writes $\delta_i$ to the residual stream and then remove it, so that the intermediate steam contains $\mathbf{h}_i + \delta_i$ and the final stream is $\mathbf{h}_i$.*

*Proof.* Since the input to the layer that computes $\delta_i$ is preserved, we can simply repeat it twice and flip signs so that the second layer writes $-\delta_i$. This guarantees that the residual stream after the first layer is $\mathbf{h}_i + \delta_i$ and the residual stream after the second layer is $\mathbf{h}_i + \delta_i - \delta_i = \mathbf{h}_i$. $\qquad\square$

### A.2 COMPUTING POSITION OFFSETS

It will be useful to show how a transformer can compute the position index of the previous token.

**Lemma 4.** *Assume a transformer stores $\mathbb{1}[i = 0]$ and $\mathbb{1}[i < k]$ in the residual stream. Then, with 1 layer, it possible to add $\phi(i - k)$ in the residual stream at indices $i \geq k$.*

*Proof.* We construct two attention heads. The first is uniform with value $\mathbb{1}[j = 0]$, and thus computes $1/i$. The second is uniform with value $\mathbb{1}[j \geq k]$, and thus computes $(i - k)/i$. We then use a feedforward layer to compute $\phi((i - k)/i, 1/i) = \phi(i - k)$ and store it in the residual stream. $\qquad\square$

The precondition that we can identify the initial token (cf. Merrill & Sabharwal, 2024) is easy to meet with any natural representation of position, including $1/i$ or $\phi(i)$, as we can simply compare the position representation against some constant.

We assume that the positional encodings used by the model allow detecting the initial token (Merrill & Sabharwal, 2024). One way to enable this would simply be to add a beginning-of-sequence token, although most position embeddings should also enable it directly.

### A.3 EQUALITY CHECKS

We show how to perform an equality check between two scalars and store the output as a boolean.

**Lemma 5.** *Given two scalars $x, y$ computable by attention heads or stored in the residual stream, we can use a single transformer layer to write $\mathbb{1}[x = y]$ in the residual stream. Furthermore, a second layer can be used to clear all intermediate values.*

*Proof.* After computing $x, y$ in a self-attention layer, we write $x - y$ to a temporary cell in the residual stream. The feedforward sublayer reads $\sigma_1 = \text{sgn}(x - y)$, computes $z = 1 - \text{ReLU}(\sigma_1) - \text{ReLU}(-\sigma_1)$, and writes $z$ to the residual stream.

The next transformer layer then recomputes $y - x$ and adds it to the intermediate memory cell, which sets it back to 0. Thus, the output is correct and intermediate memory is cleared. $\qquad\square$

## B GRAPH CONNECTIVITY PROOF

**Theorem 2.** *There exists an $(17, 2, 1)$-universal transformer $T$ with both causal and unmasked heads that, when unrolled $\lceil \log_2 n \rceil$ times, solves the connectivity problem on (directed or undirected) graphs over $n$ vertices: given as input the $n \times n$ adjacency matrix of a graph $G$, $n^3$ padding tokens, and $s, t \in \{1, \ldots n\}$ in unary notation, $T$ determines whether $G$ has a path from vertex $s$ to vertex $t$.*

*Proof.* We will prove this for directed graphs, as an undirected edge between two vertices can be equivalently represented as two directed edges between those vertices. Let $G$ be a directed graph over $n$ vertices. Let $A \in \{0, 1\}^{n \times n}$ be $G$'s adjacency matrix: for $i, j \in \{1, \ldots, n\}$, $A_{i,j}$ is 1 if $G$ has an edge from $i$ to $j$, and 0 otherwise.

The idea is to use the first $n^2$ tokens of the transformer to construct binary predicates $B_\ell(i, j)$ for $\ell \in \{0, 1, \ldots, \lceil \log n \rceil\}$ capturing whether $G$ has a path of length at most $2^\ell$ from $i$ to $j$. To this end, the transformer will use the $n^3$ padding tokens to also construct intermediate ternary predicates

$C_\ell(i, k, j)$ for $\ell \in \{1, \ldots, \lceil \log n \rceil\}$ capturing whether $G$ has paths of length at most $2^{\ell-1}$ from $i$ to $k$ and from $k$ to $j$. These two series of predicates are computed from each other iteratively:

$$B_0(i, j) \iff A(i, j) \vee i = j \tag{1}$$

$$C_{\ell+1}(i, k, j) \iff B_\ell(i, k) \wedge B_\ell(k, j) \tag{2}$$

$$B_{\ell+1}(i, j) \iff \exists k \text{ s.t. } C_{\ell+1}(i, k, j) \tag{3}$$

We first argue that $B_{\lceil \log n \rceil}(i, j) = 1$ if and only if $G$ has a path from $i$ to $j$. Clearly, there is such a path if and only if there is a "simple path" of length at most $n$ from $i$ to $j$. To this end, we argue by induction over $\ell$ that $B_\ell(i, j) = 1$ if an only if $G$ has a path of length at most $2^\ell$ from $i$ to $j$. For the base case of $\ell = 0$, by construction, $B_0(i, j) = 1$ if and only if either $i = j$ (which we treat as a path of length 0) or $A_{i,j} = 1$ (i.e., there is a direct edge from $i$ to $j$). Thus, $B_\ell(i, j) = 1$ if and only if there is a path of length at most $2^0 = 1$ from $i$ to $j$. Now suppose the claim holds for $B_\ell(i, j)$. By construction, $C_{\ell+1}(i, k, j) = 1$ if and only if $B_\ell(i, k) = B_\ell(k, j) = 1$, which by induction means there are paths of length at most $2^\ell$ from $i$ to $k$ and from $k$ to $j$, which in turn implies that there is a path of length at most $2 \cdot 2^\ell = 2^{\ell+1}$ from $i$ to $j$ (through $k$). Furthermore, note conversely that *if* there is a path of length at most $2^{\ell+1}$ from $i$ to $j$, then there must exist a "mid-point" $k$ in this path such that there are paths of length at most $2^\ell$ from $i$ to $k$ and from $k$ to $j$, i.e., $C_{\ell+1}(i, k, j) = 1$ for *some* $k$. This is precisely what the definition of $B_{\ell+1}(i, j)$ captures: it is 1 if and only if there exists a $k$ such that $C_{\ell+1}(i, k, j) = 1$, which, as argued above, holds if and only if there is a path of length at most $2^{\ell+1}$ from $i$ to $j$. This completes the inductive step.

We next describe how the transformer operationalizes the computation of predicates $B_\ell$ and $C_\ell$. The input to the transformer is the adjacency matrix $A$ represented using $n^2$ tokens from $\{0, 1\}$, followed by $n^3$ padding tokens $\square$, and finally the source and target nodes $s, t \in \{1, \ldots, n\}$ represented in unary notation using special tokens $a$ and $b$:

$$A_{1,1} \ldots A_{1,n} \; A_{2,1} \ldots A_{2,n} \; \ldots \ldots \; A_{n,1} \ldots A_{n,n} \; \underbrace{\square \ldots \ldots \ldots \square}_{n^3} \; \underbrace{a \ldots \ldots a}_{s} \; \underbrace{b \ldots \ldots b}_{t}$$

Let $N = n^2 + n^3 + s + t$, the length of the input to the transformer. The first $n^2$ token positions will be used to compute predicates $B_\ell$, while the next $n^3$ token positions will be used for predicates $C_\ell$.

**Initial Layers.** The transformer starts off by using layer 1 to store $1/N, n, n^2, s$, and $t$ in the residual stream at every position, as follows. The layer uses one head with uniform attention and with value 1 only at the first token (recall that the position embedding is assumed to separate 1 from other positions). This head computes $1/N$ and the layer adds $\psi(1/N)$ to the residual stream. Note that the input tokens in the first set of $n^2$ positions, namely 0 and 1, are distinct from tokens in the rest of the input. The layer, at every position, uses a second head with uniform attention, and with value 1 at tokens in $\{0, 1\}$ and value 0 at all other tokens. This head computes $n^2/N$. The layer now adds $\psi(n^2/N, 1/N)$, where $\psi(a, b)$ is defined as the (unnormalized) vector $\langle a, b, -a, -b \rangle$. When these coordinates are later read from the residual stream via masked pre-norm, they will get normalized and one would obtain $\phi(n^2/N, 1/N) = \phi(n^2)$. Thus, future layers will have access to $\phi(n^2)$ through the residual stream. The layer similarly uses three additional heads to compute $n^3/N$, $s/N$, and $t/N$. From the latter two values, it computes $\psi(s/N, 1/N)$ and $\psi(t/N, 1/N)$ and adds them to the residual stream; as discussed above, these can be read in future layers as $\phi(s/N, 1/N) = \phi(s)$ and $\phi(t/N, 1/N) = \phi(t)$. Finally, the layer computes $\psi(n^3/N, n^2/N)$ and adds it to the residual stream. Again, this will be available to future layers as $\phi(n^3/N, n^2/N) = \phi(n)$.

The transformer uses the next 15 layers to compute and store in the residual stream the semantic "coordinates" of each of the first $n^2 + n^3$ token position as follows. For each of the first $n^2$ positions $p = in + j$ with $1 \le p \le n^2$, it uses Lemma 1 (7 layers) with $a_i$ set to $p$ and $m$ set $n$ in order to add $\phi(i)$ and $\phi(j)$ to the residual stream at position $p$. In parallel, for each of the next $n^3$ positions $p = n^2 + (in^3 + kn + j)$ with $n^2 + 1 \le p \le n^2 + n^3$, it uses Lemma 1 with $a_i$ set to $p$ and $m$ set $n$ in order to add $\phi((i + 1)n + k)$ and $\phi(j)$ to the residual stream. It then uses the lemma again (7 more layers), this time with $a_i$ set to $(i + 1)n + k$ and $m$ again set to $n$, to add $\phi(i + 1)$ and $\phi(k)$ to the residual stream. Lastly, it uses Lemma 4 applied to $\phi(i + 1)$ to add $\phi(i)$ to the residual stream.

Layer 17 of the transformer computes the predicate $B_0(i, j)$ at the first $n^2$ token positions as follows. At position $p = in + j$, it uses Lemma 5 to compute $\mathbb{I}(\phi(A(i, j) = \phi(1))$ and $\mathbb{I}(\phi(i) = \phi(j))$; note

that $\phi(A(i,j))$, $\phi(i)$, and $\phi(j)$ are available in the residual stream at position $p$. It then uses a feedforward layer to output 1 if both of these are 1, and output 0 otherwise. This is precisely the intended value of $B_0(i,j)$. The sublayer then adds $B_0(i,j)$ to the residual stream. The layer also adds to the residual stream the value 1, which will be used to initialize the boolean that controls layer alternation in the repeated layers as discussed next.

**Repeating Layers.** The next set of layers alternates between computing the $C_\ell$ and the $B_\ell$ predicates for $\ell \in \{1, \ldots, \lceil \log n \rceil\}$. To implement this, each position $i$ at layer updates in the residual stream the value of a single boolean $r$ computed as follows. $r$ is initially set to 1 at layer 8. Each repeating layer retrieves $r$ from the residual stream and adds $1-r$ to the same coordinate in the residual stream. The net effect is that the value of $r$ alternates between 1 and 0 at the repeating layers. The transformer uses this to alternate between the computation of the $C_\ell$ and the $B_\ell$ predicates.

For $\ell \in \{1, \ldots, \lceil \log n \rceil\}$, layer $(2\ell - 1) + 8$ of the transformer computes the predicate $C_\ell(i,k,j)$ at the set of $n^3$ (padding) positions $p = n^2 + in^2 + kn + j$, as follows. It uses two heads, one with query $\langle \phi(i), \phi(k) \rangle$ and the other with query $\langle \phi(k), \phi(j) \rangle$. The keys in the first $n^2$ positions $q = i'n + j'$ are set to $\langle \phi(i'), \phi(j') \rangle$, and the values are set to $B_{\ell-1}(i', j')$. The two heads thus attend solely to positions with coordinates $(i,k)$ and $(k,j)$, respectively, and retrieve boolean values $B_{\ell-1}(i,k)$ and $B_{\ell-1}(k,j)$, respectively, stored there in the previous layer. The layer then uses Lemma 5 to compute $\mathbb{I}(B_{\ell-1}(i,k) = 1)$ and $\mathbb{I}(B_{\ell-1}(k,j) = 1)$, and uses a feedforward layer to output 1 if both of these checks pass, and output 0 otherwise. This is precisely the intended value of $C_\ell(i,k,j)$. If $\ell > 1$, the layer replaces the value $C_{\ell-1}(i,k,j)$ stored previously in the residual stream with the new boolean value $C_\ell(i,k,j)$ by adding $C_\ell(i,k,j) - C_{\ell-1}(i,k,j)$ to the same coordinates of the residual stream. If $\ell = 1$, it simply adds $C_\ell(i,k,j)$ to the residual stream.

For $\ell \in \{1, \ldots, \lceil \log n \rceil\}$, layer $2\ell + 8$ computes the predicate $B_\ell(i,j)$ at the first $n^2$ position $p = in + j$, as follows. It uses a head with query $\langle \phi(i), \phi(j) \rangle$. The keys in the second set of $n^3$ positions $q = n^2 + i'n^2 + k'n + j'$ are set to $\langle \phi(i'), \phi(j') \rangle$ (recall that $\phi(i')$ and $\phi(j')$ are available in the residual stream at $q$) and the corresponding values are set to the boolean $C_\ell(i', k', j')$, stored previously in the residual stream. The head thus attends uniformly to the $n$ padding positions that have coordinates $(i, k', j)$ for various choices of $k'$. It computes the average of their values, which equals $h = \frac{1}{n} \sum_{k'=1}^{n} C_\ell(i, k', j)$ as well as $1/(2n)$ using an additional head. We observe that $h \geq 1/n$ if there *exists* a $k'$ such that $C_\ell(i, k', j) = 1$, and $h = 0$ otherwise. These conditions correspond precisely to $B_\ell(i,j)$ being 1 and 0, respectively. We compute $h - 1/(2n)$ and store it in the residual stream. Similar to the proof of Lemma 5, the feedforward layer reads $\sigma = \mathrm{sgn}(h - 1/(2n))$, computes $z = (1 + \mathrm{ReLU}(\sigma))/2$, and writes $z$ to the residual stream. The value $z$ is precisely the desired $B_\ell(i,j)$ as $\sigma$ is 1 when $h \geq 1/n$ and 0 when $h = 0$. As in Lemma 5, the intermediate value $h - 1/(2n)$ written to the residual stream can be recomputed and reset in the next layer. As before, the transformer replaces the value $B_{\ell-1}(i,j)$ stored previously in the residual stream with the newly computed value $B_\ell(i,j)$ by adding $\psi(B_\ell(i,j) - B_{\ell-1}(i,j))$ to the stream at the same coordinates.

**Final Layers.** Finally, in layer $2\lceil \log n \rceil + 18$, the final token uses a head that attends with query $\langle \phi(s), \phi(t) \rangle$ corresponding to the source and target nodes $s$ and $t$ mentioned in the input; recall that $\phi(s)$ and $\phi(t)$ are available in the residual stream. The keys in the first $n^2$ positions $p = in + j$ are, as before, set to $\langle \phi(i), \phi(j) \rangle$, and the values are set to $B_{\lceil \log n \rceil}(i,j)$ retrieved from the residual stream. The head thus attends solely to the position with coordinates $(s,t)$, and retrieves and outputs the value $B_{\lceil \log n \rceil}(s,t)$. This value, as argued earlier, is 1 if and only if $G$ has a path from $s$ to $t$. $\quad\square$

## C  Proofs for Width Scaling and Chain of Thought Claims

**Theorem 3.** *Consider a transformer with fixed depth whose width (model dimension) grows as a polynomial of $n$ and whose weights on input length $n$ (to accomodate growing width) are computable in $\mathsf{L}$. Then this transformer can be simulated in $\mathsf{L}$-uniform $\mathsf{TC}^0$.*

*Proof.* By assumption, we can construct an $\mathsf{L}$-uniform $\mathsf{TC}^0$ circuit family in which the transformer weights for sequence length $n$ are hardcoded as constants. Next, we can apply standard arguments (Merrill et al., 2022b; Merrill & Sabharwal, 2023a;b) to show that the self-attention and feedforward

sublayers can both be simulated by constant-depth threshold circuits, and the size remains polynomial (though a larger polynomial). Thus, any function computable by a constant-depth, polynomial-width transformer is in L-uniform $\mathsf{TC}^0$. □

**Theorem 4.** *Any language recognized by a transformer with $O(\log n)$ steps of chain of thought (cf. Merrill & Sabharwal, 2024) is in* $\mathsf{TC}^0$.

*Proof.* The high-level idea is that a polynomial-size circuit can enumerate all possible $O(\log n)$-length chains of thought. Then, in parallel for each chain of thought, we construct a threshold circuit that simulates a transformer (Merrill & Sabharwal, 2023a) on the input concatenated with the chain of thought, outputting the transformer's next token. We then select the chain of thought in which all simulated outputs match the correct next token and output its final answer. The overall circuit has constant depth, polynomial size, and can be shown to be L-uniform. Thus, any function computable by a transformer with $O(\log n)$ chain of thought is in $\mathsf{TC}^0$. □

