# OpenReview forum: "A Little Depth Goes a Long Way: the Expressive Power of Log-Depth Transformers"
_ICLR.cc/2025/Conference — Submitted to ICLR 2025_

### Official Review · Reviewer_w9m8 · 2024-10-21

**Soundness:** 2
**Presentation:** 2
**Contribution:** 2
**Rating:** 6
**Confidence:** 4

**Summary:**

This paper investigates log-depth Transformers, demonstrating that their capabilities exceed those of constant-depth Transformers ($\mathsf{TC}^0$) and constant-depth Transformers with $O(\log n)$ chain-of-thought steps. Through theoretical constructions, the authors show that a log-depth Transformer can effectively solve the regular language recognition problem and the graph connectivity problem, both of which fall outside the expressive power of constant-depth Transformers with polynomial size.

**Strengths:**

The authors utilize saturated attention rather than hard attention, aligning more closely with practical applications. Additionally, they consider the effects of layer normalization.

**Weaknesses:**

1. The motivation for log-depth Transformers lacks sufficient conviction. Given the bounded context assumption in line 50, it would be more appropriate to treat all elements as constant. If bounded context and model sizes are to be discussed carefully, as indicated in lines 76-78 regarding the maximum problem size of graph connectivity with fixed depth $d$, the coefficient is crucial, while the result of $2^{O(d)}$ remains unclear.

2. The results presented lack clarity. In the main theorems (Theorems 1 and 2), only the model depth of $O(\log n)$ is specified, omitting other critical configurations such as the required model width and the number of attention heads.

3. Additional clarification is needed regarding the results in relation to prior work. Specifically, Lemma 3 of [1] demonstrates that a Transformer with depth $2d$ maps $\langle C,x\rangle$ to the circuit value $C(x)$, where $C$ is a threshold circuit with depth $d$. Since regular language recognition and the graph connectivity problem both belong to $\mathsf{TC}^1$, this lemma could already suggest comparable results in this paper.

4. The absence of experimental results to validate the theoretical findings is a significant gap. Empirical verification of the solid lines in Figure 1 is also essential.

[1] William Merrill and Ashish Sabharwal. The parallelism tradeoff: Limitations of log-precision transformers. https://arxiv.org/abs/2207.00729.

**Questions:**

1. What are the other configurations (e.g., model width, maximum number of attention heads, etc.) for the Transformers in the main theorems, specifically Theorems 1 and 2?

2. Can you provide more clarification of your results in relation to prior work, such as [1]? (Refer to weakness 3)

3. Can you conduct experiments on the two tasks considered in the paper, varying the model depth, model width, and the number of attention heads to empirically evaluate how these factors affect performance?

[1] William Merrill and Ashish Sabharwal. The parallelism tradeoff: Limitations of log-precision transformers. https://arxiv.org/abs/2207.00729.

---

> ### Author Response · Authors · 2024-11-22
> **Response to Reviewer w9m8**
>
> Thanks for your review! We appreciate your detailed technical comments and questions, and we will incorporate your feedback to improve the paper.
>
> > W1: The **motivation for log-depth** Transformers lacks sufficient conviction. Given the bounded context assumption in line 50, it would be more appropriate to treat all elements as constant. If bounded context and model sizes are to be discussed carefully, as indicated in lines 76-78 regarding the maximum problem size of graph connectivity with fixed depth d, the coefficient is crucial, while the result of 2^O(d) remains unclear.
>
> If we understand correctly, your concern is that when considering the bounded context case, it would be appropriate to treat *all* architectural parameters (depth, width, number of heads, …) as constants (w.r.t. input length $n$). We agree – given a specific input size bound $n’$, there is indeed a fixed $d’$, a fixed $w’$, a fixed number of heads, etc., that suffice to solve the task.
>
> Our theoretical and newly reported empirical results reveal what values of $(d, w)$ suffice (for the considered reasoning problems) for any desired context bound $n$. In other words, if one is only interested in solving, say, the state tracking task for inputs up to some bounded length $n$, then depth $d(n) = O(\log n)$ suffices with a small constant factor (theoretically 7, empirically less than 5), while width must scale exponentially as $w(n) = \exp(O(n))$.
>
> Does this help address your concern?
>
> > W2: The results presented lack **clarity**. In the main theorems (Theorems 1 and 2), only the model depth of O(log⁡ n) is specified, omitting other critical configurations such as the required model width and the number of attention heads.
>
> We will change the theorem statements to clarify that the model width and number of attention heads are $O(1)$ w.r.t. $n$ in our constructions in Theorem 1 and 2 – depth is the only parameter that can grow (minimally) with $n$.
>
> > W3: Additional clarification is needed regarding the results in **relation to prior work**. Specifically, Lemma 3 of [1] demonstrates that a Transformer with depth 2d maps ⟨C,x⟩ to the circuit value C(x), where C is a threshold circuit with depth d. Since regular language recognition and the graph connectivity problem both belong to TC1, this lemma could already suggest comparable results in this paper.
>
> This is an interesting connection! However, the argument does not quite go through.
> If the transformer is allowed to receive an *input-length dependent* prompt of size poly(n) before the input x, then indeed you’re right that, following from [1], there is an input-dependent prompt C(|x|) (namely a log-depth threshold circuit that implements regular language recognition or graph connectivity for inputs of size |x|) such that a log-depth circuit will map ⟨C(|x|),x⟩ to y.
>
> However, there are a couple problems with using this idea to try to prove Theorem 1 or 2. First, the mapping from x to a representation of C(|x|) is actually quite complicated, and it’s unclear whether a transformer can compute it. Further, transformer gets as input just x, and it cannot “prepend” additional tokens C(|x|) to be read by later layers. In other words, while regular language recognition and graph connectivity are reducible in some sense to the circuit value problem (CVP), transformers cannot directly implement the reduction. Thus, it does not directly follow from the CVP construction in [1] that log-depth transformers can express automata or graph connectivity. We thus obtain these results through other means in Theorems 1 and 2.
>
> To the extent that space allows, we will add a discussion along these lines to the paper!
>
> > W4: The absence of **experimental results** to validate the theoretical findings is a significant gap. Empirical verification of the solid lines in Figure 1 is also essential.
>
> Regarding the lack of experimental results, we have added extensive experimental results on the A5 state tracking task that confirm the theoretical predictions around depth and width (see the *general response*). We will replace Figure 1 with a figure showing these empirical results. We hope these experiments, in addition to the comments above, have addressed your main concerns and believe they have improved the quality of the paper.
>
> > Q1: What are the **other configurations** (e.g., model width, maximum number of attention heads, etc.) for the Transformers in the main theorems, specifically Theorems 1 and 2?
>
> All other configurations in the model architecture are constants; we will clarify this (see response above to W2).
>
> > Q2: Can you provide more clarification of your results in relation to prior work, such as [1]? (Refer to weakness 3)
>
> Please see the answer to W3 above.
>
> > Q3: Can you conduct experiments on the two tasks considered in the paper…
>
> Yes! Please see the answer to W4 above.

---

> > ### Comment · Reviewer_w9m8 · 2024-11-22
> >
> > Thank you for your response!
> >
> > Although this paper offers scaling insights for Transformers in tasks such as regular language recognition and graph connectivity, demonstrating that scaling depth is more effective than scaling width (or even applying CoT), the chosen tasks are relatively specific.  It remains unclear whether these insights can generalize to more broader scenarios.  Could you provide additional discussion to address this point?
> >
> > I also encourage the authors to upload a revised version of the paper, clarifying key concepts and incorporating the new experimental section.  Including the experimental results discussed in the general response as part of the main text would further strengthen the manuscript.

---

> > > ### Author Response · Authors · 2024-11-22
> > > **PDF Updated with Experiments**
> > >
> > > > I also encourage the authors to upload a revised version of the paper, clarifying key concepts and incorporating the new experimental section.
> > >
> > > We have updated the PDF to include the new experimental results! New text is (temporarily) highlighted in blue for your convenience. We have also clarified some details in the definitions section around the definition of language recognition for transformers and different types of masking with transformers. In Section 8, we have also added some further discussion about the other problems where depth scaling may be less effective compared to recognizing regular languages, which constitute interesting directions for future work.
> > >
> > > > It remains unclear whether these insights can generalize to more broader scenarios. Could you provide additional discussion to address this point?
> > >
> > > Our negative results about width and CoT are general: they show that any reasoning problem outside TC0 (which contains many types of sequential reasoning) cannot be solved without width *exploding exponentially* or CoT steps growing *more than logarithmically*. In contrast, our positive result in Theorem 1 only shows that, for the specific problem of regular language recognition (outside TC0), growing depth $\sim \log n$ enables recognizing strings of length $n$.
> > >
> > > In future work, plan to extend our results to better understand how much depth is required for other problems harder than state tracking, including CFG recognition and evaluating compositional formulas and circuits. This could lead to a general understanding of the relationship between model depth and different types of reasoning.
> > > 1. **Empirically:** We plan to do this by adapting our experimental setup here to harder problems like the ones mentioned above. We have not done this yet because we are limited by computational constraints (the experiments with A5 took 1000 GPU hours on their own).
> > > 2. **Theoretically:** we have ideas about how to produce a tight characterization of the expressive power of log-depth transformers building on the specific constructions from this paper. However, this proof is quite involved and still a work in progress, so it lies outside the scope of the current project.
> > >
> > > As mentioned above, we have added some discussion along these lines to Section 8 and plan to further clarify these points in Section 6.

---

> > > > ### Comment · Reviewer_w9m8 · 2024-11-23
> > > >
> > > > Thank you again for your response. While Theorems 3 and 4 have already been established in prior work [1] (see Figure 10 for an illustration), I expect to see further exploration of the expressive power of $O(\log n)$-depth Transformers. This paper focuses on positive results for two specific problems, which limits the generalizability of the findings.
> > > >
> > > > I'm happy to see the revision of the paper. As a reminder, the main text should be between 6 and 10 pages for ICLR 2025, whereas the current version exceeds this limit.
> > > >
> > > > [1] Chain of Thought Empowers Transformers to Solve Inherently Serial Problems, https://arxiv.org/abs/2402.12875.

---

> ### Author Response · Authors · 2024-11-25
> **Length updated and other changes**
>
> Thanks for the reminder about the page limit. We've now updated the submission PDF to fit within 10 pages. We've also cited Figure 10 from Li et al in Section 6 and given them credit (while also maintaining our proofs in the appendix for completeness).
>
> While we agree that we focus on the problems of regular language recognition and graph connectivity (and plan to extend to more general results about classes of problems in the future), we note that analyzing specific problems often paves the way towards general results, and also that these are quite important problems. Regular language recognition, in particular, formalizes many types of real-world state tracking, as has been noted in previous work:
> * https://arxiv.org/abs/2404.08819
> * https://arxiv.org/abs/2210.10749
>
> We appreciate your suggestions and believe our revised manuscript is substantially improved to address your original weaknesses (clarifying definitions, connection to prior work, and adding experiments). Regarding the motivation for log depth, we have also added the following paragraph to Section 2:
>
>  > An $(s,r,t)$-transformer unrolled to some fixed depth can be viewed as a uniform special case of a fixed-depth transformer. Thus, constructions of dynamic-depth transformers (depth $d(n)$ for inputs of length $n$) imply that, given any bounded context length $N$, there also exists a fixed-depth transformer with depth $d(N)$ for the task at hand. The fact that this can be done with a looped transformer with dynamic depth is, in fact, a stronger condition that shows the construction is uniform, which is formally important as non-uniform models of computation can have very strong and unrealistic power. In this way, our results about looped transformers will provide insights about standard, non-looped transformers with bounded context lengths.

---

> > ### Comment · Reviewer_w9m8 · 2024-11-25
> >
> > Thanks for your response. I will increase my score.

---

### Official Review · Reviewer_ok53 · 2024-10-30

**Soundness:** 3
**Presentation:** 2
**Contribution:** 3
**Rating:** 5
**Confidence:** 3

**Summary:**

The paper addresses the expressive capacities of transformers where
the depth grows logarithmically with the length of the inputs. Specifically, the paper seeks to
contribute to three questions:
- Can fixed-depth transformers handle "hard" problems when restricted to bounded input lengths?
- Does logarithmic depth scaling offer greater expressive abilities compared to a fixed depth?
- What practical advantages do log-depth transformers or input-dependent depth offer?

Informally, the paper's primary theoretical contributions regarding the "universal transformer" model,
which is characterised by a basic encoder architecture with certain layers repeated a number of times
dependent on the input length, are as follows:
- (Theorem 1) For each regular language $L$, there exists a universal transformer, with logarithmic repetition,
that recognises $L$.
- (Theorem 2) There exists a universal transformer, with logarithmic repetition, that decides whether,
given a graph $G$ and vertices $s$ and $t$, there is a path from $s$ to $t$.
- (Theorems 3 & 4) Placing different classes of transformers (fixed depth or CoT) in $\text{TC}^0$.

**Strengths:**

- The three areas that the paper aims to contribute to are both relevant and intriguing. Particularly,
the assumption of bounded input lengths, which is closer to practical settings, deserves thorough investigation.
- Setting aside the notational and definitional ambiguities discussed below, the paper supports all
theoretical claims with proofs, which appear to be sufficient.
- The related work, specifically regarding the expressive capabilities of different transformer models,
is well chosen. I concur that this paper contributes significantly to these domains.

**Weaknesses:**

To begin, I will summarize the main issue I encounter with this paper.
As mentioned earlier, the contributions the paper aims to provide are intriguing, and I also think they are partially achieved.
However, the current form of the paper makes it challenging to grasp the significance of these contributions and how they compare to existing results. This difficulty arises primarily from a lack of clear definitions, confusing informal statements, and a disconnect between the theoretical contributions and their proposed implications.

I will go into detail below, loosely ordering the weaknesses from most to least significant:
- I am not convinced that the paper sufficiently addresses the first and third contributions it aims to make. The main results focus on transformers with dynamic depth, leaving me unclear about the implications for fixed-depth models. Lines 76 to 83 seem to mention this in the context of an experiment, but it is only briefly referenced and lacks further development throughout the paper. Additionally, the explanation of the third contribution in Lines 86 to 9, which states, "scaling depth logarithmically as a computational resource more efficiently expands the expressive power of transformers compared to scaling width...", appears to be an overinterpretation based solely on the results of Theorems 1 & 2 versus Theorem 3. If this claim is to be maintained, the paper needs to provide more detailed justification for such a bold statement, which is currently absent.
- The definition of the (s, r, t)-universal transformer model lacks clarity and detail. While I understand the authors' intent to be succinct, even those familiar with related research might struggle to comprehend the model's specifics. A thorough understanding is crucial for comparing the expressiveness of different transformer models. Here are some points related to the definition in Section 2:
    - There seems to be a confusion in terminology regarding the "semantics" of universal transformers, which are described as sequence-to-value functions. However, on line 103, they are referred to as a decoder, which is typically associated with sequence-to-sequence transformations.
    - The definition of masking within this model is unclear. The phrase "... add a learned mask vector that can select specific dimensions of the residual stream ..." is ambiguous and needs clarification. This issue is particularly important because, for example, in Lemma 1, the term "...causally masked attention..." is mentioned, but its meaning is not well defined.
    - The notation $L^l$ is ambiguous. Does it signify layer $l$ or the function it performs? Additionally, the notation $r$-layer $(l-s) \mod r$ is not defined
    - The definition of averaging-hard attention using a limits construction and an exponential function diverges from the common use of hardmax in related works cited within the paper. Are there differences that justify this choice?
    - Section 2.2 appears disjointed and complicates the definition. The first sentence raises multiple questions, particularly on the necessity of "memory management."
- The presentation of Theorem 2 is somewhat misleading. The problem addressed is not the "connectivity problem" in the traditional sense, which involves deciding whether all nodes are interconnected by some path. Instead, it is the "reachability problem," which focuses on deciding if there is a path connecting specific nodes.
- The role of Theorem 4 is not clearly articulated. The theorem is referenced with
(Anonymous, p.c.), which is somewhat unconventional. Is this result established within this paper?
It seems there is a proof provided in the appendix, so the rationale for this citation is unclear.
If the theorem is not original to this work, clarification is necessary regarding its inclusion.
- The abstract begins with the statement: "Most analysis of transformer expressivity treats the depth (number of layers) of a model
as a fixed constant, ...". I find this statement problematic. It seems that the authors are referring to research focusing on formal language theory, which aims to understand which classes of languages specific classes of transformers can recognize.
While individual transformer models have a fixed depth, this is right, the class of models does not need to be constrained in this way. This is an example of an informal assertion that leads to confusion.
- Lines 44 to 46 include the statement, "This is analogous to how regular expressions cannot express all context-free languages, but one can write regular expressions that capture fragments of a context-free language." The paper uses this analogy to imply that transformers generally perform effectively on shorter input lengths. This statement is somewhat perplexing because the point about regular expressions is quite basic; if there is additional nuance or significance, such statements require further clarification.

**Questions:**

I would appreciate it if you could address the weaknesses I highlighted earlier. Specifically, I am interested in the following:

- Could you provide additional clarification on your contributions regarding the first question, particularly on how you contribute to understanding the limits of the expressiveness of fixed-depth transformers with bounded inputs?
- Could you offer a more detailed and rigorous explanation of the concepts discussed in Section 2.2?
- Am I correct in identifying a potential confusion between decoder-only and encoder-only models, or is there a detail I might have overlooked?
- What is the precise method of masking you employ within your model?

I'm keen to resolve these issues as I believe your work holds significant value. However, the points mentioned above currently obscure its impact.

---

> ### Comment · Reviewer_ok53 · 2024-11-21
>
> I encourage the authors to engage in a discussion with me, even if you can only address some of my questions at this stage.
>
> As the rebuttal phase approaches its conclusion and considering the various points I have raised, I believe an early discussion would be beneficial.

---

> > ### Author Response · Authors · 2024-11-21
> > **Re: Official Comment by Reviewer ok53**
> >
> > We very much appreciate your nudge and encouragement, thank you!
> >
> > We have been preparing a detailed response, including extensive empirical support as asked for by multiple reviewers, and will post it here shortly.
> >
> > Best,
> > Authors

---

> > > ### Author Response · Authors · 2024-11-22
> > > **Response to Reviewer ok53 (Part 1)**
> > >
> > > Thank you for your review!
> > >
> > > > I am not convinced that the paper sufficiently addresses the first and third contributions it aims to make. The main results focus on transformers with dynamic depth, leaving me unclear about the implications for fixed-depth models.
> > >
> > > > Could you provide additional clarification on your contributions regarding the first question, particularly on how you contribute to understanding the limits of the expressiveness of **fixed-depth transformers** with bounded inputs?
> > >
> > > Regarding the connection between fixed-depth and dynamic-depth models, we note that a dynamic-depth model unrolled to some fixed depth $d$ can be viewed as a “uniform” special case of a model with fixed depth $d$. Thus, our constructions for dynamic-depth transformers (depth $d(n)$ for inputs of length $n$) imply that, given any bounded context length $n’$, it is possible to construct a *fixed-depth* transformer (with depth $d’ = d(n’)$) for the task at hand. The fact that this can be done with a looped transformer with dynamic depth is, in fact, a stronger condition that shows that the construction is not inherently non-uniform (i.e., not too strange or contrived, in a certain formal way). In this way, our results about looped transformers also provide insights about standard, non-looped transformers when operating on bounded context lengths.
> > >
> > > We also note that our new empirical results (see *general response*), which are with standard, non-looped transformers, are in line with our theoretical findings, which are with looped transformers. This again highlights the close connection between the two.
> > >
> > > > Additionally, the explanation of the **third contribution** in Lines 86 to 9, which states, "scaling depth logarithmically as a computational resource more efficiently expands the expressive power of transformers compared to scaling width...", appears to be an overinterpretation based solely on the results of Theorems 1 & 2 versus Theorem 3. If this claim is to be maintained, the paper needs to provide more detailed justification for such a bold statement, which is currently absent.
> > >
> > > We agree that the claim in the original draft is too general and will make it more precise. Concretely, within the specific context of regular language recognition, we have theoretical and empirical evidence that $O(\log n)$ layers suffice to recognize strings of length $n$, whereas the width would have to grow as $\exp(O(n))$. It is entirely possible (and likely) that for other tasks, growing width might be more effective. For example, this could be the case for knowledge-heavy tasks or tasks that can be efficiently parallelized. Our claim only applies for specific sequential tasks that are outside of TC0. We will clarify this in the discussion of Theorem 3.
> > >
> > > > Could you offer a more detailed and rigorous explanation of the concepts discussed in **Section 2.2**?
> > >
> > > We appreciate the many different suggestions you brought up to clarify the definitions in Section 2.2 (defining masking, layer notation, etc.). We plan to incorporate these changes in our revision so that the definition section is more precise and readable. Most crucially, the details around masking are quite important, which we elaborate on in response to your next question.
> > >
> > > > Am I correct in identifying a potential confusion between **decoder-only and encoder-only** models, or is there a detail I might have overlooked? What is the precise method of **masking** you employ within your model?
> > >
> > > This detail indeed got lost while editing the definitions section and theorem statements -- thank you for bringing it up! To clarify, Theorem 1 about regular language recognition holds for decoder-only transformers: that is, the transformer receives as input the input string x, attends with causally masked attention, and generates the output as a single token y. In contrast, our graph connectivity construction in Theorem 2 requires both causally masked as well as unmasked attention, which is mentioned in the proof, although this got lost in the high-level text. This is definitely a simplifying assumption that we will more carefully note.
> > >
> > > We will also point out that, using general tricks, the unmasked heads could be removed from the construction by simply adding $O(log n)$ blank tokens to the input. The graph connectivity result, as currently stated, would also go through with causally-masked attention for any graph without cycles, assuming the nodes are topologically ordered in the input.
> > >
> > > We will make sure to clarify this limitation of the graph connectivity result as well as some discussion of these finer points about how it could be extended.

---

> > > > ### Author Response · Authors · 2024-11-22
> > > > **Response to Reviewer ok53 (Part 2)**
> > > >
> > > > > The presentation of Theorem 2 is somewhat misleading. The problem addressed is not the "connectivity problem" in the traditional sense, which involves deciding whether all nodes are interconnected by some path. Instead, it is the "**reachability problem**," which focuses on deciding if there is a path connecting specific nodes.
> > > >
> > > > We refer to the problem as graph connectivity because it is often called [st-connectivity](https://en.wikipedia.org/wiki/St-connectivity) in the computational complexity literature, but you’re correct that it is equally valid, and perhaps clearer, to call it reachability. We can clarify this when we introduce the problem.
> > > >
> > > > > The role of Theorem 4 is not clearly articulated. The theorem is referenced with **(Anonymous, p.c.)**, which is somewhat unconventional. Is this result established within this paper? It seems there is a proof provided in the appendix, so the rationale for this citation is unclear.
> > > >
> > > > Thanks for this feedback - we see how this is confusing. This theorem is indeed a newly published result and we give our own proof in the appendix. We cited (Anonymous p.c.) because one source of inspiration for the proof was an insightful comment from an audience member at a workshop, and we wished to acknowledge them. However, upon further thought, this seems like an unconventional choice, and we think it would be more appropriate to simply mention the comment from the person in the acknowledgments.
> > > >
> > > > > The **abstract** begins with the statement: "Most analysis of transformer expressivity treats the depth (number of layers) of a model as a fixed constant, ...". I find this statement problematic. It seems that the authors are referring to research focusing on formal language theory, which aims to understand which classes of languages specific classes of transformers can recognize. While individual transformer models have a fixed depth, this is right, the class of models does not need to be constrained in this way. This is an example of an informal assertion that leads to confusion.
> > > >
> > > > Indeed, we are referring to the typical computational model analyzed in the literature that analyzes transformers in terms of formal language theory. We were not attempting to make any kind of formal statement here, but rather simply point out a common assumption in a prominent line of prior work. We will clarify the sentence to say “Most analysis of transformer expressivity in terms of formal language theory treats…”, which hopefully satisfies your concern. We are happy to hear alternative suggestions. Our main goal is to highlight that this is the first formal analysis that does not consider the depth to be a fixed constant (w.r.t. input size).
> > > >
> > > > > Lines 44 to 46 include the statement, "This is **analogous to how regular expressions** cannot express all context-free languages, but one can write regular expressions that capture fragments of a context-free language." The paper uses this analogy to imply that transformers generally perform effectively on shorter input lengths. This statement is somewhat perplexing because the point about regular expressions is quite basic; if there is additional nuance or significance, such statements require further clarification.
> > > >
> > > > We think this sentence, and the analogy, will be clearer if stated as “This is analogous to how regular expressions cannot express all context-free languages, but given a maximum sequence length, it is possible to write a regular expression that matches any desired context-free grammar up to that sequence length.” The idea is that, if the size of a less expressive model can grow with sequence length, it is possible for it to approximate more expressive models, and one can analyze the computational resources (e.g., number of states or, for transformers, depth or width) that is required to do so. This sentence was simply meant as an analogy to help readers better understand our research question and results. If you find it unhelpful or distracting, we would also be happy to delete it.

---

> > > > ### Comment · Reviewer_ok53 · 2024-11-23
> > > > **response to questions part 1**
> > > >
> > > > >  Thus, our constructions for dynamic-depth transformers
> > > >  for inputs of length imply that, given any bounded context length, it is possible to construct a fixed-depth transformer for the task at hand.
> > > >
> > > > > The fact that this can be done with a looped transformer with dynamic depth is, in fact, a stronger condition that shows that the construction is not inherently non-uniform
> > > >
> > > > Allright, this makes sense and having understood this, your investigations are far more intriguing.
> > > >
> > > > However, can you point me to where in the paper this is made clear? It can very likely be the case that I just missed (or forgot) a paragraph where this is stated as nicely as you done above.
> > > >
> > > > > We appreciate the many different suggestions you brought up to clarify the definitions in Section 2.2
> > > >
> > > > As I am myself working in a somewhat similar area, I feel this matter of rigorously defining the models under consideration is of utmost importance. By now, there are many papers related to expressive power of "transformers" in one way or another. But, I feel there are equally many confusions about what result applies to which "transformer model" by now.
> > > > This makes the comparison of expressiveness results very difficult. So thanks for trying to be careful here.
> > > >
> > > > However, my main point of criticism was  specifically related to 2.2 and this "memory management" stuff. I have a slight feeling of what you are getting at, but its informal nature is in a strong contrast to the otherwise formal Section 2 and to be honest not really clear. Can you explain its importance? Can you clarify it anyhow?

---

> ### Comment · Reviewer_ok53 · 2024-11-23
>
> > because it is often called st-connectivity in the computational complexity literature,
>
> Okay, I see this connection. For me, reachability is the natural one. Do as you wish, but I would recommend mention it somewhere as I think I am not the only one whose more used to reachability.
>
> > ..., which hopefully satisfies your concern. [...] We are happy to hear alternative suggestions. [...] Our main goal is to highlight that this is the first formal analysis that does not consider the depth to be a fixed constant (w.r.t. input size).
>
> The first part sounds like I should actually be named "reviewer pIckY53". Jokes aside, I am aware that this is only an informal statement and should be treated as such. But, especially when concerned with formal research like you are, these informal statements should be handled with extra care. To me, this sentence conveys "most expressiveness research focuses on models with a bounded depth". I see there can be other interpretations, but you probably agree that its problematic as soon as there is more than one. However, this was only a minor point.
>
> > This sentence was simply meant as an analogy to help readers better understand our research question and results. If you find it unhelpful or distracting, we would also be happy to delete it.
>
> I appreciate your mission here. But I do not think that this analogy helps very much.
>
> ---
>
> So far, I feel that this submission touches the threshold of acceptance. I see its worth and am willed to increase my rating, but please answer the follow-up questions I raised first, because these are important to me.

---

> > ### Author Response · Authors · 2024-11-25
> > **Response to questions**
> >
> > > However, can you point me to where in the paper this is made clear? It can very likely be the case that I just missed (or forgot) a paragraph where this is stated as nicely as you done above.
> >
> > Reading back over the original manuscript, you're right that this point was not made as explicitly as it should have been. Previously, it was referred to more implicitly in the introduction and conclusion, but we should definitely make the connection more clear. We have therefore updated the part of the introduction that introduces our contributions to make the connection to fixed-depth transformers more explicit. We have also added a paragraph after the looped transformer definition in the top of Section 2 that summarizes our words to you in more detail. Together, these two changes should help address the issue you raised.
> >
> > > However, my main point of criticism was specifically related to 2.2 and this "memory management" stuff. I have a slight feeling of what you are getting at, but its informal nature is in a strong contrast to the otherwise formal Section 2 and to be honest not really clear. Can you explain its importance? Can you clarify it anyhow?
> >
> > In the main text, we mainly focus on the memory management problem: in short, it is that a looped transformer will write many times to the same cell in memory, and previously written values may interfere with our ability to retrieve later values that are written there. In particular, we want to be able to have something like "temporary variables" in the repeated layers of the transformers: cells in the residual stream we can allocate dynamically within each iteration to store some temporary values.
> >
> > Our tool to achieve this is Lemma 3. The idea of this lemma is that it is possible for transformers to "undo" the computation of a single layer. That is, given hidden state h, if we can add f(h) to the residual stream with one layer, it is possible to revert the residual stream back to h with another layer. Intuitively, this is useful because it allows us to create temporary variables that store some intermediate computation and which we know will have been free'ed when the next iteration of the repeated layers starts. Currently, the details of this Lemma are buried in the appendix. We will edit the main text to make this intuition more clear.
> >
> > Finally, some quick comments on your other points. Regarding st-connectivity, we can clarify that the problem could also be referred to as reachability. We'll also clarify the sentence you pointed out in the abstract. We've also decided to cut the analogy to regular vs. context-free languages, as it will help free up space we could spend in better ways.

---

> > > ### Comment · Reviewer_ok53 · 2024-11-27
> > >
> > > Thank you for your prompt reply! I was expecting a clarification regarding the "memory management" aspects of your paper, but your response unfortunately accentuated my general concerns:
> > >
> > > > in short, it is that a looped transformer will write many times to the same cell in memory, and previously written values may interfere with our ability to retrieve later values that are written there.
> > >
> > > I'm sorry, but I am struggling to grasp the formal meaning here. Initially, the mention of "cell in memory" might suggest hardware-related issues. After reviewing your paper briefly, I note that this terminology isn't clearly defined.
> > >
> > > This led me to re-examine parts of your paper. I recalled that Appendix A provides a definition of "storing values," which I believe provides an answer: *We say that a scalar z is stored in the residual stream if some set of four indices contains either $\psi(z)$ or $\varphi(z)$.*
> > > I assume "indices" refers to positions in the input word? How does a position contain a value? Do you mean the residual stream concerning these indices? I am uncertain, and I find it perplexing how mingled formal definitions are with informal statements.
> > >
> > > The same goes partly for the definition of the residual stream $\boldsymbol{h}_i$: I am still not completely sure what the definition in 117-119 imply.
> > >
> > > > Our tool to achieve this is Lemma 3.
> > >
> > > Lemma 3 states: *Assume there exists a single transformer layer that writes an update $\delta_i$ to the residual stream $\boldsymbol{h}_i$ using indices at which $\delta_i$ is 0. Then there is a block of two transformer layers that writes $\delta_i$ to the residual stream and then removes it, so that the intermediate stream contains $\boldsymbol{h}_i + \delta_i$ and the final stream is $\boldsymbol{h}_i$.*
> > >
> > >
> > > The immediate questions I have are:
> > > - What does "[...] $\boldsymbol{h}_i$ using indices at which $\delta_i$ is 0" mean?
> > > - What does "writes $\delta_i$ to the residual stream" imply?
> > > - Does "and then removes it" mean subtract? If I am not mistaken, this is not mentioned in Appendix A.
> > >
> > > Please do not misunderstand me, I am not endeavouring to nitpick. I can imagine that, with considerable time investment, one could study your paper thoroughly and make educated guesses about these informal statements. However, the current style doesn't facilitate a swift and clear understanding of the formal results, arguments, and their correctness.
> > >
> > > Regarding the proof of Lemma 3: it appears trivial, which again puzzles me as to why it requires so much elaboration. It might not be as trivial as it seems, but at this point, I can only speculate.
> > >
> > > ---
> > >
> > > At this point I feel that my initial evaluation of the paper is justified, although I in between thought that I see it above the threshold: The contributions are of interest and I would really like to see them published, but the current style of the paper obscures them. This is mainly due to informal definitions and a lack of mathematically precise arguments. I do not really see how this can be resolved without a major revision.

---

> > > > ### Author Response · Authors · 2024-11-27
> > > >
> > > > > Please do not misunderstand me, I am not endeavouring to nitpick.
> > > >
> > > > On the contrary, we appreciate your engagement and attention to detail! We will respond to each of your questions in detail below and incorporate these clarifications in revision to make the section more precise.
> > > >
> > > > ## Indices and Storage
> > > >
> > > > > I assume "indices" refers to positions in the input word?
> > > >
> > > > In this context, indices mean different elements of the vector $\mathbf h_i \in \mathbb R^d$. That is, we have some quantity $z_i$ at each position $i$, and, for each $i$, we want the residual stream $\mathbf h_i$ to encode $\phi(i)$.
> > > >
> > > > > How does a position contain a value? Do you mean the residual stream concerning these indices?
> > > >
> > > > Correct. We use $\mathbf h_i$ to indicate the residual stream. From our original PDF:
> > > >
> > > > > When a general scalar $z$ needs to be preserved, we store it as a 4-dimensional vector. Let $\psi(z) = \langle z, 1, -z, -1 \rangle$ be its *unnormalized representation* and the corresponding $0$-centered unit vector $\phi(z) = \psi(z) / \sqrt{z^2 + 1}$ be its *normalized representation*. We say that a scalar $z$ is *stored in the residual stream* if some set of four indices contain either $\psi(z)$ or $\phi(z)$.
> > > >
> > > > We will update to make this clearer. Basically, 4 dimensions of $\mathbf h_i$ will be used to store $\phi(z_i) \in \mathbb R^4$. Hopefully the previous answer also makes this clearer.
> > > >
> > > > ## Lemma 3 Clarification
> > > >
> > > > > What does "[...] $\mathbf h_i$ using indices at which $\delta_i$ is 0" mean?
> > > >
> > > > We just mean that the original contents of the residual stream $\mathbf h_i$ and the update vector $\delta_i$ should partition the dimensions of the residual stream, i.e.,
> > > >
> > > > $$
> > > > \mathbf h_{i+1} = \langle \mathbf h_i, \vec 0 \rangle + \langle \vec 0, \delta_i \rangle = \langle \mathbf h_i, \delta_i \rangle
> > > > $$
> > > >
> > > > This was not clearly worded and in fact "at which $\delta_i$ is 0" was a typo (vs. "$\mathbf h_i$ is 0"). We will clarify the statement of the Lemma towards the clarification above.
> > > >
> > > > ## Writing and Removing
> > > >
> > > > > What does "writes to the residual stream" imply?
> > > >
> > > > We quoted above the paragraph that defines "stores" for the residual stream, but, indeed, you're write that "writes" is not formally defined. In Lemma 3 and 4, when we say "writes $x$", we simply mean that the layer ouputs the vector $x$ and then, by definition of the transformer architecture, $x$ gets added to the residual stream. We will clarify this by simply rewording "writes $x$" to "outputs $x$ and adds it to the residual stream" where appropriate.
> > > >
> > > > > Does "and then removes it" mean subtract? If I am not mistaken, this is not mentioned in Appendix A.
> > > >
> > > > This is correct and analogous to above. More precisely, "removes $x$" means "outputs $-x$ and adds it to the residual stream". The statement of Lemma 3 (and proof) make this precise: the first layer residual stream is $\mathbf h_i + \delta_i $ and the second layer residual stream is $\mathbf h_i + \delta_i - \delta_i = \mathbf h_i$. We will clarify this.

---

### Official Review · Reviewer_oAHH · 2024-11-01

**Soundness:** 3
**Presentation:** 2
**Contribution:** 1
**Rating:** 3
**Confidence:** 3

**Summary:**

The work shows that two reasoning problems where standard transformer struggled to solve can be solved by a looped transformer. The work also shows that modular arithmetic with small modulus can be solved with the same architecture. Authors finally showed that scaling depth is more efficient than width or adding CoT intermediate steps.

**Strengths:**

S1. Authors showed that looped transformer (non-constant depth architecture) can solve problems more efficiently than CoT or adding width. This is quite interesting for theoretical transformer foundation, although the impact is pretty limited for current constant-depth architecture, given it's not clear and easy to scale large training for looped transformer.

S2. The findings could inform more efficient model scaling strategies, maybe combining the effect of CoT with log n depth.

S3. The paper is well written and all relevant lemmas are well explained.

**Weaknesses:**

W1. Major weakness is that the theoretical results lack experimental validation on the tasks author suggested, limiting real world applicability. Also, dismissing CoT with a couple of lines seems unfair, given the potential showed by CoT on arithmetic and graph related problems, above all.

W2. Given the memory management issue, implementing dynamical depth is not clear and also it's not clear if a looped transformer can be indeed __trained__ to solve the two mentioned problems.

W3. The paper is very intricate and a bit more explanation may be needed to understand this work. Also the work is not really self contained.

W4. Although theorem 3 and 4 are the core part of the authors work, they are not well justified.

**Questions:**

Q1. Given the two differences from the standard transformer architecture and the fact that Lemma 1 holds because of the masked pre-norm, do you think that the results hold for standard transformer architecture?

Q2. Would it be possible to add a more informal introduction to let other people read and fully understand the work without looking at prior work?

Q3. Did you also see a similar law for other architectures (like https://arxiv.org/abs/2405.06394 or other constant transformer-like model)?

---

> ### Author Response · Authors · 2024-11-22
> **Response to Reviewer oAHH (Part 1)**
>
> Thanks for your review!
>
> > W1: Major weakness is that the theoretical results lack **experimental validation** on the tasks author suggested, limiting real world applicability.
>
> We agree that experiments would improve the original paper, especially because the theory makes testable predictions. We have added extensive experiments that confirm the predicted theoretical relationship between effective context length for state tracking and depth and width (see the *general response*). We believe that these results strengthen our argument and its applicability and would be curious to hear if they address your concern.
>
> > W1: Also, dismissing **CoT** with a couple of lines seems unfair, given the potential showed by CoT on arithmetic and graph related problems, above all.
>
> We agree that the comparison to chain of thought in section 6 could be made more nuanced. Specifically, we will clarify that the context in which we have found growing depth is more efficient than chain of thought is only for specific reasoning problems. For other sorts of reasoning problems, it is feasible that chain of thought could be a better computational fit than growing depth, and there might be other concerns (simplicity, trainability, etc.) under which chain of thought is more appealing than looping layers. We will include this discussion.
>
> > W3: The paper is **very intricate** and a bit more explanation may be needed …
>
> We plan to clarify the transformer definitions and theorem statements as discussed in responses to other reviewers. This will help simplify the proof structure in the main results (Theorems 1-2). Regarding Theorems 3 and 4, we view these as simple results that help contextualize the main results in Theorems 1-2 (by comparing the effect of depth to width or CoT steps). We will add more discussion to these sections to make the role of these results clearer. Regarding justification, note that these theorems do have proofs that were moved to the appendix for space. Regarding self-containedness, if there are specific technical tools that are not adequately introduced beyond those we have mentioned, please let us know and we would be happy to incorporate additional background in the appendix.
>
> > W2: Given the memory management issue, implementing dynamical depth is not clear and also it's not clear if a looped transformer can be indeed **trained** to solve the two mentioned problems.
>
> The constructions in Theorem 1 and 2 show how, in principle, it is possible to get around the memory management issue with hand-crafted constructions. You are right that a question remains about whether such constructions are learnable in practice. The experiments we have added (see general response) get at this question to an extent: we find that, in practice, transformers with depth ~ $\log n$ can learn to express state tracking over $n$ items. However, our empirical results focused on non-looped transformers: it would be interesting to replicate our analysis with looped transformers but we cannot do this currently due to the significant computational cost. We will acknowledge this as an interesting direction for future work in the paper.
>
> > W4: Although theorem 3 and 4 are the core part of the authors work, they are **not well justified**.
>
> We actually view Theorems 1 and 2 as the main contributions of this paper, while Theorem 3 and 4 are meant to contextualize our main results about depth scaling against other width scaling or chain of thought. Regarding the justification of Theorems 3 and 4, we note that we have included proofs for the theorems, but deferred them to the appendix due to space. We agree with the reviewer that more space could be used to explain the significance of these theorems. In particular, we will add some text explaining the importance of Theorem 3: since regular language recognition and graph connectivity are outside TC0, this result suggests width must scale exponentially w.r.t. $n$ to solve these problems up to bounded length $n$. Our new experiments confirm this theoretical prediction.

---

> > ### Author Response · Authors · 2024-11-22
> > **Response to Reviewer oAHH (Part 2)**
> >
> > > Q1. Given the two differences from the standard transformer architecture and the fact that Lemma 1 holds because of the **masked pre-norm**, do you think that the results hold for standard transformer architecture?
> >
> > We believe so. We haven’t been able to extend the theory to standard pre-norm, but our new empirical results (with standard pre-norm transformers) provide strong evidence that a construction is achievable with standard pre-norm.
> >
> > > Q2. Would it be possible to add a **more informal** introduction to let other people read and fully understand the work without looking at prior work?
> >
> > Thanks, we agree this would be valuable. We plan to add more high-level discussion to contextualize Theorems 1-4, the new experimental results, and the overall takeaway from the theory and experiments. To make space, we will likely have to relegate some more proofs to the appendix.
> >
> > > Q3. Did you also see a similar law for **other architectures** (like https://arxiv.org/abs/2405.06394 or other constant transformer-like model)?
> >
> > Interesting question. Due to the differences to transformers, we’re not sure how our results would translate to memory mosaics. Extending our analysis of depth scaling to memory mosaics or other architectures like state-space models would be a very interesting direction, especially because past work has already considered the state-tracking capabilities of (constant-depth) state-space models.

---

> > > ### Author Response · Authors · 2024-11-27
> > > **Request for comment**
> > >
> > > Thanks again or your review. We're eager to hear whether our response, new experiments, and writing changes have changed your assessment of the paper.
> > >
> > > In particular, we hope the newly added experiments address your main concern of missing empirical support (W1). We have also taken into account W2-W4 both in our responses above and in our edits to the submission PDF, and would be curious to hear your opinion.

---

### Official Review · Reviewer_zS8A · 2024-11-03

**Soundness:** 4
**Presentation:** 4
**Contribution:** 4
**Rating:** 8
**Confidence:** 2

**Summary:**

The paper explores the computational expressivity of transformers when their depth scales logarithmically with input context length. Prior analyses typically assume a fixed transformer depth, which limits the model’s ability to solve certain tasks as context length increases. This work, however, argues that transformers can still solve certain problems up to a bounded input length, even if they cannot handle them for arbitrarily large inputs. By scaling depth logarithmically with the input length, transformers can effectively simulate finite automata and solve graph connectivity problems, which are critical for tasks involving multi-step reasoning and state tracking. These findings suggest that only a modest, logarithmic increase in depth is required to address such tasks, offering a path for efficient model scaling and highlighting the benefits of dynamic depth adjustments over simply expanding model width or employing chain-of-thought reasoning.

**Strengths:**

1. **Innovative Depth Scaling Insight**: This paper shifts the focus from fixed-depth transformers to a dynamic, log-depth scaling, addressing some limitations of traditional models in handling extended contexts. This perspective broadens our understanding of transformers’ potential expressivity.

2. **Rigorous Theoretical Foundation**: The paper provides clear mathematical proofs and complexity analyses that validate the advantages of log-depth scaling for specific tasks, particularly regular language recognition and graph connectivity. This rigor strengthens the work’s contributions to understanding transformers’ computational capacity.

3. **Comprehensive Comparison with Other Scaling Approaches**: The authors examine depth scaling in comparison to width scaling and chain-of-thought methods, demonstrating that logarithmic depth growth is more computationally efficient and effective for reasoning tasks, especially for state tracking and pathfinding in bounded contexts.

**Weaknesses:**

1. **Limited Consideration of Complementary Methods**: The paper promotes depth scaling over width or chain-of-thought expansion but could benefit from a more nuanced discussion of scenarios where those methods may still offer advantages or may complement log-depth scaling.

2.  **Lack of Experimental Validation**: Although the theoretical findings are compelling, the paper would be stronger with empirical experiments demonstrating the practical performance of log-depth transformers on real-world tasks and quantitatively comparing their efficiency and effectiveness with other models.

**Questions:**

1. Are there specific conditions or types of tasks where width expansion or chain-of-thought reasoning may still be advantageous compared to log-depth scaling?

2. Do you have plans to empirically validate these theoretical results on real-world datasets? If so, what metrics or benchmarks would you prioritize to assess the practical benefits of log-depth scaling?

---

> ### Author Response · Authors · 2024-11-22
> **Response to Reviewer zS8A**
>
> Thanks for your review! We appreciate that you found our results about depth scaling insightful.
>
> Regarding the consideration of **complementary methods**, we agree that the discussion in the original draft could be expanded and made more nuanced. For instance, we will clarify that we have found a previously unknown advantage of scaling depth over width for solving *reasoning problems* outside TC0 (such as state tracking), but there are likely other kinds of tasks (e.g., those that are *knowledge intensive* or that involve large amounts of *parallel search*) where width may be more beneficial than depth. Additionally, scaling depth comes with a cost in terms of training time compared to scaling width, which we will mention explicitly. We will also contextualize our results with empirical papers comparing depth and width scaling (e.g., [Petty et al., 2024](https://aclanthology.org/2024.naacl-long.402.pdf)). Regarding the comparison with chain of thought, we agree that there is also nuance to add. Concretely, we will clarify that we have only found depth scaling to be more efficient than chain of thought on the specific problem of state tracking. Further, we will mention that, while transformers with log precision and polynomial chain of thought steps converge to P (*all* polynomial time solvable problems; [Merrill & Sabharwal, 2024](https://arxiv.org/pdf/2310.07923)), a result like this could not go through for depth scaling because the memory would be bounded at $n^2 \log n$. We believe clarifying these comparisons and expanding the discussion will improve the value of the paper for readers. Thank you, again, for bringing this up.
>
> We also agree that the original submission could be improved by adding **experiments**, especially since our theory makes testable claims about the relationship between depth and effective context length for certain reasoning tasks. We have therefore run extensive experiments to empirically verify that the context length at which transformers can perform state tracking does scale as expected with depth and width: see the *general response* for more details. We believe these results are a strong complement to the theoretical results and enhance the overall story and potential impact of the paper. Thank you for suggesting this!

---

### Author Response · Authors · 2024-11-22
**General Response**

We appreciate the reviewers’ valuable comments. In addition to answering many specific points below, we have undertaken extensive experiments to test the theoretical predictions made by our paper. We believe these new results strengthen the paper significantly and are eager to hear the reviewers’ impressions.

## Experimental Results

As several reviewers pointed out, our paper makes empirically testable predictions about the relationship between a model’s depth (and width) and the effective context length for key reasoning problems outside TC0. Namely: is it the case that recognizing regular languages over strings of length $n$ requires depth ~ $\log n$ and width $\sim \exp(O(n))$, and, if so, what are the coefficients?

We ran an extensive set of experiments to address these questions, training models of different depths and widths on the A5 state tracking task ([Merrill et al., 2024](https://arxiv.org/abs/2404.08819)) that is a canonical testbed for hard regular language recognition (Theorem 1). The input to the task is a sequence of permutation items in A5 (the alternating permutation group over 5 elements), and the label at each token is the cumulative product of previous permutations.

We train several standard (not looped) transformers of varying depth $d$ and width $w$ on ~100 million sequences of this form (this took ~1000 GPU hours). After each model is trained, we use a test set to measure $n^*$, the maximum context length $n$ at which the model achieved 95%+ performance on the A5 state tracking task. We trained several seeds with the same $(d,w)$ setting and aggregate these results across all models with the same $(d,w)$ by taking the best-performing (max $n^*$) model. We then plot $n^*$ as a function of $d$ and $w$, evaluating if the predicted theoretical relationships hold (via an $r^2$ statistic) and the slope of the fit.

The results are shown in these **[anonymously hosted figures](https://postimg.cc/gallery/DmfkFPy)**. When varying depth (first plot), there is a very strong correlation between $d$ (x-axis) and $\log n$ (y-axis, log scale), the effective context length till which it can solve problems with high accuracy, with $r^2=0.93$. When varying width (second plot), there is an even stronger correlation between $\log w$ (x-axis, log scale) and $n^*$ (y-axis), with $r^2=0.98$. These results provide strong empirical support for our theoretical predictions that:

Increasing depth logarithmically as $d \sim \log n$ will enable state tracking up to length $n$ (Theorem 1),
but that width $w$ must increase *exponentially* with $n$ to enable state tracking over $n$ tokens (Theorem 3)

Interestingly, this empirical analysis also gives us a functional form that allows us to *quantitatively predict* the depth or width required for state tracking up to context length $n$. Concretely, the slope of the $d \sim \log n$ regression is 4.80, which is somewhat better than the slope of 7 derived in Theorem 1. As our analysis in Theorem 1 was not fully tight, we believe the construction could be improved to derive a tighter slope closer to what is actually learned in practice, which we will make a note of in the discussion.

We believe these empirical results are quite exciting as they take a step towards closing the gap between theoretical constructions and an actionable understanding of how much depth is required in transformers to solve different kinds of reasoning problems.

---

> ### Author Response · Authors · 2024-11-22
> **Experimental Section Added to PDF**
>
> We have updated the PDF to include the new experimental results discussed above. The new text is (temporarily) highlighted in blue to make it easier for you to find it. The lack of experiments was a point of concern in the original reviews, so we would appreciate hearing whether these experiments, which verify the theorized relationships between depth, width, and context length for regular language recognition, change your assessment of the paper.

---

### Meta-Review · Area_Chair_Lz4w · 2024-12-23

**Metareview:**

This paper formalizes and investigates the kinds of computation a transformer can perform on inputs of bounded length. This is accomplished by studying the expressive capacities of transformers in which the depth grows logarithmically with the length of the inputs. Theoretical results show that unlike a constant depth transformer,  a log-depth transformer can recognize regular languages and solve the graph connectivity problem. Experimental results are provided to test the predictions derived from the theoretical analysis.

**Additional Comments On Reviewer Discussion:**

The reviewers raised a variety of questions and weaknesses about the paper, and I thank the reviewer and authors for engaging in a discussion to improve the paper.  During the discussion the authors provided experiments to test the theoretical predictions made by the paper. The presence of these experiments improves the paper, however this is a substantial change to the paper that was made after the reviews were submitted. Consequently, it has been challenging to judge this paper as a whole. The consensus from the reviewer discussion was that the paper seemed rushed, there were still some concerns from the reviews that had not been addressed, and that certain technical details were unclear and ill-defined. The reviewers agreed that the paper had substantial promise, as it identifies a new and interesting problem and provides relatively complete theoretical results. However, we are recommending rejection as addressing the remaining concerns and making all technical details precise and rigorous would make the paper more impactful.

---

### Decision · Program_Chairs · 2025-01-22

Reject